# DIFFUSING TO THE TOP: BOOST GRAPH NEURAL NETWORKS WITH MINIMAL HYPERPARAMETER TUNING

**Lequan Lin** *†
University of Sydney

**Dai Shi***
University of Sydney

**Andi Han**
Riken AIP

**Zhiyong Wang**
University of Sydney

**Junbin Gao**
University of Sydney

## ABSTRACT

Graph Neural Networks (GNNs) are proficient in graph representation learning and achieve promising performance on versatile tasks such as node classification and link prediction. Usually, a comprehensive hyperparameter tuning is essential for fully unlocking GNN's top performance, especially for complicated tasks such as node classification on large graphs and long-range graphs. This is usually associated with high computational and time costs and careful design of appropriate search spaces. This work introduces a graph-conditioned latent diffusion framework (GNN-Diff) to generate high-performing GNNs based on the model checkpoints of sub-optimal hyperparameters selected by a light-tuning coarse search. We validate our method through 166 experiments across four graph tasks: node classification on small, large, and long-range graphs, as well as link prediction. Our experiments involve 10 classic and state-of-the-art target models and 20 publicly available datasets. The results consistently demonstrate that GNN-Diff: (1) boosts the performance of GNNs with efficient hyperparameter tuning; and (2) presents high stability and generalizability on unseen data across multiple generation runs. The code is available at `https://github.com/lequanlin/GNN-Diff`.

## 1 INTRODUCTION

Graph neural networks (GNNs) are deep learning models that excel in capturing relationships within graph-structured data, making them highly effective for a variety of graph tasks such as node classification and link prediction (Wu et al., 2020; Zhou et al., 2020). Although GNNs achieve good performance on simple graph tasks without sophisticated hyperparameter tuning (Kipf & Welling, 2017), fully unlocking their potential to achieve top performance or handle complicated graph tasks still requires extensive hyperparameter tuning (Luo et al., 2024; Tönshoff et al., 2024). This is often associated with high computational and time costs and careful design of appropriate search spaces. For example, in Figure 1, we show that the average test accuracy of GCN (Kipf & Welling, 2017) for node classification typically increases with the hyperparameter search space. Nevertheless, it

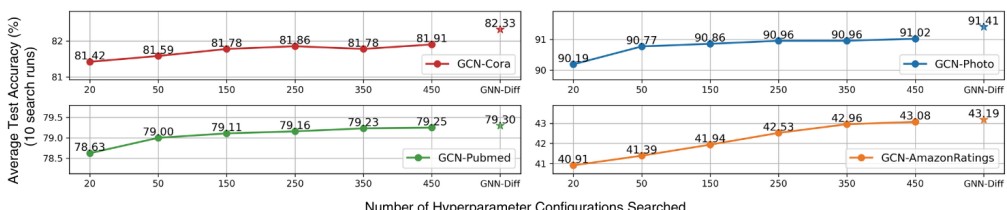

Figure 1: Effect of hyperparameter search space on GCN performance for node classification. The test accuracy is averaged over 10 search runs. We also show the results of our method, GNN-Diff.

---

*Equal contribution. Lequan Lin is the corresponding author. ✉ `lequan.lin@sydney.edu.au`
†In memory of MeiMei 🐕 , whose love and companionship will always be remembered.

Figure 2: GNNs with parameters found by (a) grid search with the full search space, (b) random search with the sub-search space, (c) coarse search with the minimal search space, and (d) GNN-Diff generation based on sub-optimal hyperparameters selected by the coarse search. Top performance may be achieved by a large search space at a cost of time, or generated with GNN-Diff efficiently.

may cost over $20\times$ of tuning time to gain an accuracy increase of less than $1\%$. This motivates us to develop a method that boosts the performance of GNNs while minimizing search space and tuning time. Furthermore, this method will be particularly beneficial for complex tasks such as node classification in large and long-range graphs (Hu et al., 2020a; Dwivedi et al., 2022), where hyperparameter tuning often plays a critical role.

In this paper, we propose a graph-conditioned latent diffusion framework (GNN-Diff) to generate high-performing GNN parameters with sub-optimal hyperparameters selected by a coarse search, which is a random search with a relevantly small search space (Figure 2). Our method assumes that the performance of GNNs is primarily determined by the quality of the learned parameters. In addition, high-quality parameters exist in the underlying parameter population produced by a sub-optimal but sufficiently good configuration, which can be found by the coarse search. Based on our experiments, the coarse search space only needs to be $10\%$ of the grid search space for most graph tasks to find a promising configuration. The parameter generation is fulfilled by a latent denoising diffusion probabilistic model (Ho et al., 2020; Rombach et al., 2022). While parameter generation have been extensively studied in previous research (Erkoç et al., 2023; Peebles et al., 2022; Schürholt et al., 2022; Soro et al., 2024; Wang et al., 2024), the intrinsic relationship between data characteristics and network parameters remains underexplored. To fill this gap, we adopt a task-oriented graph feature encoder to integrate data and graph structural information as a condition for GNN parameter generation. Lastly, it is noteworthy that GNN-Diff does not require additional hyperparameter tuning for parameter generation, reducing tuning costs to only those incurred by the coarse search.

Results from **166 experiments** across 4 graph tasks, 10 target models, and 20 datasets consistently show that GNN-Diff: (1) efficiently boosts GNN performance with minimal hyperparameter tuning; and (2) demonstrates high stability and generalizability over multiple generation runs on unseen data.

## 2 PRELIMINARIES

### 2.1 GRAPH NEURAL NETWORKS

Graph Neural Networks (GNNs) are deep-learning architectures designed for graph data. Typically, there are two types of GNNs: spatial GNNs that aggregate neighboring nodes with message passing (Hamilton et al., 2019; Kipf & Welling, 2017; Gasteiger et al., 2019) and spectral GNNs developed from the graph spectral theory (Defferrard et al., 2016; Lin & Gao, 2023; Zou et al., 2023). The key component of GNNs is the convolutional layer, which usually consists of graph convolution, linear transformation, and non-linear activation function.

We denote an undirected graph with $N$ nodes as $\mathcal{G}\{\mathcal{V}, \mathcal{E}, \mathbf{A}\}$, where $\mathcal{V}$ and $\mathcal{E}$ are the node set and the edge set, respectively, and $\mathbf{A} \in \mathbb{R}^{N \times N}$ is the adjacency matrix containing information of relationships. $\mathbf{A}$ can be either weighted or unweighted, normalized or unnormalized. The graph signals are stored in a matrix $\mathbf{X} \in \mathbb{R}^{N \times D_f}$, where $D_f$ is the number of features. Then, the convolutional layer in GCN can be written as $\sigma(\mathbf{AXW})$, involving graph convolution by node aggregation with the adjacency $\mathbf{A}$, a feature linear transformation operator $\mathbf{W}$, and the activation $\sigma$. Usually, learnable parameters of GNNs exist in the linear operator and sometimes in the parameterized graph convolution as well (Defferrard et al., 2016; Zheng et al., 2021).

## 2.2 Latent Diffusion Models

Diffusion models generate a step-by-step denoising process that recovers data in the target distribution from random white noises. A typical diffusion model adopts a pair of forward-backward Markov chains, where the forward chain perturbs the observed samples eventually to white noises, and then the backward chain learns how to remove the noises and recover the original data. Let $\mathbf{w}_0 \sim q_{\mathbf{w}}(\mathbf{w}_0)$ be the original data (e.g., vectorized network parameters) from the target distribution $q_{\mathbf{w}}(\mathbf{w}_0)$. Then, the forward-backward chains are formulated as follows.

**Forward chain.** For diffusion steps $t = 0, 1, ..., T$, Gaussian noises $\epsilon \sim \mathcal{N}(\mathbf{0}, \mathbf{I})$ are injected to $\mathbf{w}_t$ until $q_{\mathbf{w}}(\mathbf{w}_T) := \int q_{\mathbf{w}}(\mathbf{w}_T|\mathbf{w}_0) q_{\mathbf{w}}(\mathbf{w}_0) d\mathbf{w}_0 \approx \mathcal{N}(\mathbf{w}_T; \mathbf{0}, \mathbf{I})$. By the Markov property, one may jump to any diffusion steps via $\mathbf{w}_t = \sqrt{\widetilde{\alpha}_t}\mathbf{w}_0 + \sqrt{1 - \widetilde{\alpha}_t}\epsilon$, where $\widetilde{\alpha}_t = \prod_{i=1}^{t}(1 - \beta_i)$ with $\beta = \{\beta_1, \beta_2, ..., \beta_T\}$ is a pre-defined noise schedule.

**Backward chain.** The backward chain removes noises from $\mathbf{w}_T$ gradually via a backward transition kernel $q_{\mathbf{w}}(\mathbf{w}_{t-1}|\mathbf{w}_t)$, which is usually approximated by a neural network $p_\theta(\mathbf{w}_{t-1}|\mathbf{w}_t)$ with learnable parameters $\theta$. In this paper, we adopt the framework of denoising diffusion probabilistic models (DDPMs), which alternatively uses a denoising network $\epsilon_\theta$ to predict the noise injected at each diffusion step with the loss function

$$\mathcal{L}_{\text{DDPM}} = \mathbb{E}_{t, \mathbf{w}_0 \sim q_{\mathbf{w}}(\mathbf{w}_0), \epsilon \sim \mathcal{N}(\mathbf{0}, \mathbf{I})} \left\| \epsilon - \epsilon_\theta \left( \sqrt{\widetilde{\alpha}_t}\mathbf{w}_0 + \sqrt{1 - \widetilde{\alpha}_t}\epsilon, t \right) \right\|^2. \tag{1}$$

**Latent diffusion for parameter generation.** Generating GNN parameters directly with diffusion models may lead to slow training and long inference time, especially when the model is heavily-parameterized. As a common solution to such problems, latent diffusion models (LDM) (Rombach et al., 2022; Soro et al., 2024; Wang et al., 2024) first learn a parameter autoencoder (PAE) to convert the target parameters $\mathbf{w}_0 \in \mathbb{R}^{D_w}$ to a low-dimensional latent space $\mathbf{z}_0 = \text{P-Encoder}(\mathbf{w}_0) \in \mathbb{R}^{D_p}$ with $D_p \ll D_w$, then generate in the latent space before reconstructing the original target with a sufficiently precise decoder. The loss function of PAE is usually formulated as the mean squared error (MSE) between original parameters $\mathbf{w}_0$ and reconstructed parameters $\text{P-Decoder}(\mathbf{z}_0)$.

## 3 Related Works

This is a summary of related works, and we discuss more details in Appendix A. The related works cover three topics: network parameter generation, GNN training, and advanced methods for hyperparameter tuning. In network parameter generation, we start with hypernetwork, an early concept of predicting parameters of the target network (Ha et al., 2017; Stanley et al., 2009), followed by a review on generative models for parameter generation (Schürholt et al., 2021; 2022; Peebles et al., 2022; Wang et al., 2024; Soro et al., 2024). p-diff (Wang et al., 2024) is the most relevant method to our approach, which collects checkpoints from the training process of the target network and adopts an unconditional latent diffusion model to generate high-performing parameters. Next, in GNN training, we discuss two major approaches to boost GNN performance, including pre-training (Hu et al., 2020b;c; Lu et al., 2021) and architecture search (Gao et al., 2020; You et al., 2020). However, both approaches inevitably rely on hyperparameter tuning to succeed. Lastly, we consider some advanced methods of hyperparameter tuning, such as Bayesian optimization (Snoek et al., 2012; 2015) and coarse-to-fine search (Moshkelgosha et al., 2017; Payrosangari et al., 2020).

## 4 Graph Neural Network Diffusion (GNN-Diff)

GNN-Diff is a graph-conditioned diffusion framework that generates high-performing parameters for a target GNN to match or even surpass the results of time-consuming hyperparameter tuning with a much more efficient process. This is achieved via a pipeline of four steps: (1) input graph data, (2) parameter collection of the target GNN with a light-tuning coarse search, (3) training, and (4) inference and prediction. GNN-Diff has three trainable components: a parameter autoencoder (PAE), a graph feature encoder (GFE), and a graph-conditioned latent diffusion model (G-LDM). We provide the overview of the GNN-Diff pipeline in Figure 3. The input data and PAE have been previously discussed in Section 2, thus the rest of this section focuses on the remaining details in the GNN-Diff pipeline. The pseudo codes of training and inference algorithms are provided in Appendix B.

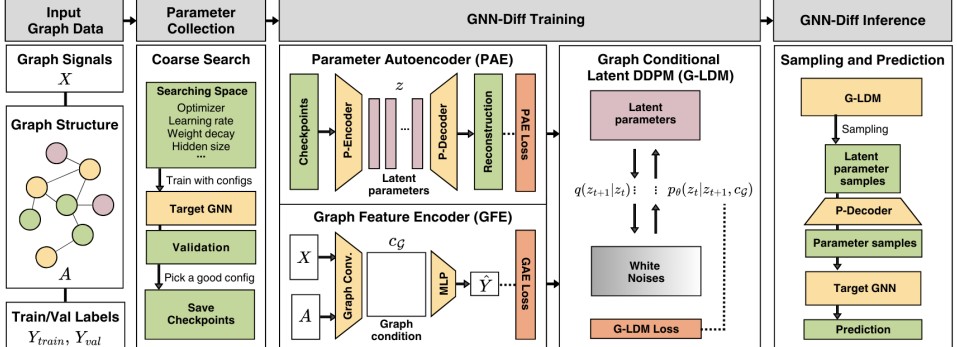

Figure 3: How GNN-Diff works for node classification. (1) Input graph data: input graph signals, adjacency matrix, and train/validation ground truth labels. (2) Parameter collection: we use a coarse search with a small search space to select an appropriate hyperparameter configuration for the target GNN, and then collect model checkpoints with this configuration. (3) Training: PAE and GFE are firstly trained to produce latent parameters and the graph condition, and then G-LDM learns how to recover latent parameters from white noises with the graph condition as a guidance. (4) Inference: after sampling latent parameters from G-LDM, GNN parameters are reconstructed with the PAE decoder and returned to the target GNN for prediction.

## 4.1 PARAMETER COLLECTION WITH COARSE SEARCH

To produce high-performing GNN parameters, we need a set of good parameter samples such that the diffusion model can generate from their underlying population distribution. Intuitively, the quality of parameter samples relies on the hyperparameter selection. Hence, we apply a coarse search with a relevantly small search space to determine the suitable configuration. Two types of hyperparameters are considered: training-related such as learning rate and weight decay, and model-related such as hidden size and the teleport probability $\alpha$ in APPNP (Gasteiger et al., 2019). We train the target model with multiple runs for each configuration. Then, we select the configuration with the best validation results on average. After the coarse search, model checkpoints trained with the selected configuration are saved as parameter samples. Parameter samples are reorganized and vectorized before they are further processed by the PAE.

## 4.2 GRAPH FEATURE ENCODER (GFE)

Previous works have discussed using datasets as conditions of parameter generation, but only for the purpose of transferability (Nava et al., 2022; Soro et al., 2024; Zhang et al., 2024). The role of specific data characteristics in parameter generation still remains underexplored. To fill this gap, GNN-Diff leverages the graph data and structural information as the generative condition. The GFE is designed to encode graph information in graph signals $\mathbf{X}$ and structure $\mathbf{A}$ to form the condition. Our experiments involve node classification and link prediction tasks, thus the GFE encoder aims at producing a graph representation that works well on the specific task. Straightforwardly, we consider adopting a GNN-based architecture for the encoder to capture and process both data and graph information. This architecture is determined by the nature of each task. For example, multiple GNN layers may be well-suited for node classification on homophilic graphs, where connected nodes are normally from the same classes, but inappropriate for heterophilic graphs, where nodes with different labels are prone to be linked (Zheng et al., 2022; Han et al., 2024). Hence, enlightened by some previous works (Shao et al., 2023; Thorpe et al., 2022; Zhu et al., 2020), we introduce a structure that is capable of handling node classification on both homophilic and heterophilic graphs. The GFE encoder (without activation) can be denoted as

$$\eta = \text{G-Encoder}(\mathbf{X}, \mathbf{A}) = \text{Concat}\left(\mathbf{A}^2\mathbf{X}\mathbf{W}_1, \mathbf{A}\mathbf{X}\mathbf{W}_2, \mathbf{X}\mathbf{W}_3\right)\mathbf{W}_4 \in \mathbb{R}^{N \times D_p}, \quad (2)$$

which is composed of the concatenation of a two-layer GCN, a one-layer GCN, and an MLP, followed by a linear transformation $\mathbf{W}_4 \in \mathbb{R}^{D_{\mathcal{G}} \times D_p}$, where $D_{\mathcal{G}}$ and $D_p$ are the dimensions of the concatenation and the latent parameter representation, respectively. Next, the GFE decoder for general node classification is a single linear layer G-Decoder$(\eta) = \eta\mathbf{W}_5 \in \mathbb{R}^{D_c}$, with $D_c$ being the

number of node classes. This ensures that graph information is fully encapsulated in the output of the encoder. Finally, the loss function is the cross entropy on node labels. We use this GFE for node classification on small homophilic and heterophilic graphs. The GFEs for node classification on large and long-range graphs and link prediction are discussed in Appendix C.1. The graph condition is constructed via mean-pooling the nodes as $\mathbf{c}_{\mathcal{G}} = \frac{1}{N}\sum_{i=1}^{N}\eta(i) \in \mathbb{R}^{D_p}$, where $\eta(i)$ is the i-th row of graph representation. The purpose of applying mean pooling on nodes is to construct an overall graph condition regardless of the graph size, which will later be combined with latent representation $\mathbf{z}_0 \in \mathbb{R}^{D_p}$ as input of diffusion denoising network.

## 4.3 GRAPH-CONDITIONED LATENT DDPM (G-LDM)

G-LDM is developed from LDM with the graph condition as a consolidated guidance on latent GNN parameter generation. The forward pass of G-LDM is the same as traditional DDPM, while the backward pass uses a graph-conditioned backward kernel to recover data from white noises. Given the latent parameters $\mathbf{z}_0$ from the PAE encoder and the graph condition $\mathbf{c}_{\mathcal{G}}$ from the GFE encoder, the corresponding G-LDM loss function is written as

$$\mathcal{L}_{\text{G-LDM}} = \mathbb{E}_{t,\mathbf{z}_0\sim q_{\mathbf{z}}(\mathbf{z}_0),\epsilon\sim\mathcal{N}(\mathbf{0},\mathbf{I})} \left\| \epsilon - \epsilon_\theta\left(\sqrt{\widetilde{\alpha}_t}\mathbf{z}_0 + \sqrt{1-\widetilde{\alpha}_t}\epsilon, t, \mathbf{c}_{\mathcal{G}}\right) \right\|^2. \tag{3}$$

For both PAE and denoising network $\epsilon_\theta$, we adopt the `Conv1d`-based architecture in (Wang et al., 2024). We include the graph condition by taking the sum of $\mathbf{z}_t$, $\mathbf{c}_{\mathcal{G}}$, and the time embedding as input to $\epsilon_\theta$. We observe in experiments that the most important factor in high-quality generation is to reduce parameter dimension sufficiently while ensuring reconstruction tightness. This can be easily done by setting the appropriate kernel size and stride for `Conv1d` layers in PAE based on the parameter size, which does not require hyperparameter tuning. We discuss more details about the PAE and $\epsilon_\theta$ architectures and how to ensure high-quality generation in Appendices C.2 and C.3.

## 4.4 INFERENCE AND PREDICTION

The last step is to obtain parameters and return them to the target GNN for prediction. G-LDM generates samples of latent parameters with the learned $\epsilon_\theta$. Then, the learned PAE decoder is applied to reconstruct parameters back to the original parameter space. These parameters are returned to the target GNN, thus we have a group of GNNs with generated parameters. Lastly, the model with the best validation performance will be used for final evaluation.

# 5 EXPERIMENTS AND DISCUSSIONS

## 5.1 EXPERIMENTAL SETUP

**Tasks and metrics.** We evaluate our method with 4 graph tasks: (1) basic node classification on small homophilic and heterophilic graphs; (2) node classification on large graphs with a huge amount of nodes; (3) node classification on long-range graphs with very deep GNNs; (4) link existence prediction on undirected graphs. We use accuracy as the metric for all tasks except for the node classification on long-range graphs. Since node classes are highly imbalanced in long-range graphs, we employ macro F1-score as the metric. We will report $100\times$ F1-scores for presentation purpose.

**Baseline methods.** We compare GNN-Diff with (1) grid search with full search space; (2) random search with 20% of full search space; (3) coarse search with 10% of full search space; (4) coarse-to-fine (C2F) that narrows down search space based on the coarse search to optimize hyperparameters.

**Datasets.** We conduct the experiments on 20 publicly available datasets. For basic node classification, we consider 6 homophilic graphs: `Cora`, `Citeseer`, and `Pubmed` (Yang et al., 2016), as well as `Computers`, `Photo`, and `CS` (Shchur et al., 2018); and 6 heterophilic graphs: `Actor` and `Wisconsin` (Pei et al., 2020), along with `Roman-Empire`, `Amazon-Ratings`, `Minesweeper`, and `Tolokers` (Platonov et al., 2023). We also test on large graphs with 89,250 to 2,339,029 nodes, including `Flickr` (Zeng et al., 2020), `Reddit` (Hamilton et al., 2019), and `OGB-arXiv` and `OGB-Products` (Hu et al., 2020a). For long-range graphs, we use `PascalVOC-SP` and `COCO-SP` (Dwivedi et al., 2022). Additionally, our link prediction experiments involve 4 datasets: `Cora` and `Citeseer` (Yang et al., 2016), as well as `Chameleon` and `Squirrel` (Rozemberczki et al., 2021).

**Target models.** 9 classic and state-of-the-art GNNs are selected as target models: GCN (Kipf & Welling, 2017), SAGE (Hamilton et al., 2019), APPNP (Gasteiger et al., 2019), GAT (Veličković et al., 2018), ChebNet Defferrard et al. (2016), H2GCN (Zhu et al., 2020), SGC (Wu et al., 2019), GRPGNN (Chien et al., 2021), and MixHop (Abu-El-Haija et al., 2019). We also include multilayer perceptron (MLP) to represent baseline performance.

**Experiment details.** All experiments are run on NVIDIA 4090 GPUs with 24GB memory. GNN-Diff is trained with a consecutive training flow of its three components (200 epochs for GFE, 9000 epochs for PAE and 6000 epochs for G-LDM). We may generate either all or part of the target model's parameters. The partial generation focuses on the last learnable layer, because earlier layers typically perform the role of feature extraction and representation learning, while the last layer is more task-specific and crucial for the final classification (Yosinski et al., 2014). For the sake of efficiency, GNN-Diff performs partial generation in all experiments. To collect partial parameters, we fix the parameters in all layers except the last layer after 190 epochs, save the checkpoint with the best validation result, and further train it with a small learning rate $5e$-4 in another 10 epochs for 10 runs. The parameters collected in each run are different due to randomness caused by dropout, but they are assumed to be from the same population, as they are obtained by slightly altering the saved checkpoint. In evaluation, we generate 100 samples and test the sample with the best validation performance. For the baseline search methods, we first train the target model with each configuration for 200 epochs over 10 runs (3 runs for some large and long-range graphs in consideration of time), then select the configuration with the best average validation performance. To ensure a fair comparison with GNN-Diff, which fine-tunes the last layer for parameter collection, we also train the target model for 190 epochs and fine-tune the last layer for 10 more epochs using the best configuration from the baseline search. We then report the better test accuracy from either the full 200-epoch training or the fine-tuning scheme.

**Reproducibility and comparability.** The random seed was set as 42 to mitigate randomness in experiments. Our results may be different from some related works such as (Luo et al., 2024; Tönshoff et al., 2024) due to different data split and model architecture. For example, we only use the first split of `Actor` and `Wisconsin` (Pei et al., 2020) because all the experiments are conducted based on one train/validation/test split. Also, the target GNNs mostly adopt simple architectures without tricks such as residual connection and layer or batch normalization.

Please refer to Appendices D, E, F, G, and H for more experiment settings and details.

Table 1: Results of basic node classification on homophilic and heterophilic graphs. We report the average test accuracy (%) and the standard deviation of models selected by validation over 10 training or generation runs. The best results are marked in **bold**.

| Datasets | Cora (Homophily) | | | | | Citeseer (Homophily) | | | | |
|---|---|---|---|---|---|---|---|---|---|---|
| Models | Grid | Random | Coarse | C2F | GNN-Diff | Grid | Random | Coarse | C2F | GNN-Diff |
| MLP | 59.41 ± 0.94 | 58.32 ± 1.21 | 58.28 ± 0.68 | 57.83 ± 1.58 | **59.47 ± 0.43** | 58.26 ± 1.04 | 57.63 ± 1.45 | 57.51 ± 1.30 | 57.63 ± 1.44 | **58.72 ± 0.84** |
| GCN | 82.04 ± 0.96 | 81.52 ± 0.71 | 81.89 ± 0.48 | 81.99 ± 0.95 | **82.33 ± 0.17** | 71.92 ± 1.10 | 72.04 ± 0.56 | 71.97 ± 0.67 | 72.13 ± 0.23 | **72.37 ± 0.29** |
| SAGE | 80.58 ± 1.04 | 80.49 ± 0.77 | 80.43 ± 0.78 | 80.43 ± 0.78 | **80.60 ± 0.15** | **70.56 ± 0.58** | 69.50 ± 0.61 | 68.81 ± 0.86 | 70.39 ± 0.91 | 70.45 ± 0.14 |
| APPNP | 81.91 ± 0.90 | 81.69 ± 0.62 | 81.47 ± 0.51 | 81.91 ± 0.90 | **82.51 ± 0.29** | 70.82 ± 1.40 | 69.87 ± 0.45 | 69.68 ± 0.63 | 69.60 ± 1.32 | **71.44 ± 0.17** |
| GAT | 81.10 ± 0.52 | 81.05 ± 0.82 | 80.13 ± 1.18 | 80.52 ± 1.29 | **81.69 ± 0.10** | 70.83 ± 0.58 | 70.81 ± 0.69 | 70.45 ± 1.13 | 69.82 ± 1.10 | **71.50 ± 0.09** |
| ChebNet | 81.83 ± 0.46 | 81.51 ± 0.85 | 81.30 ± 0.88 | 81.73 ± 1.09 | **82.05 ± 0.55** | 71.14 ± 0.13 | 71.24 ± 0.75 | 71.02 ± 1.12 | 71.14 ± 0.13 | **71.65 ± 0.27** |
| H2GCN | 82.05 ± 0.81 | 81.67 ± 0.71 | 81.63 ± 0.88 | 82.11 ± 0.72 | **82.17 ± 0.12** | 71.49 ± 0.89 | 71.30 ± 1.31 | 71.39 ± 0.19 | 71.12 ± 0.98 | **71.78 ± 0.25** |
| SGC | 81.89 ± 0.94 | 82.09 ± 0.55 | 81.60 ± 0.80 | 82.09 ± 0.55 | **82.10 ± 0.24** | 71.94 ± 0.56 | 71.82 ± 0.26 | 71.77 ± 0.23 | 71.70 ± 0.43 | **72.10 ± 0.18** |
| GPRGNN | 81.03 ± 0.65 | 81.34 ± 0.63 | 81.21 ± 0.64 | 81.41 ± 0.84 | **81.79 ± 0.20** | 70.93 ± 1.19 | 70.44 ± 0.71 | 70.05 ± 0.94 | 70.93 ± 1.19 | **71.86 ± 0.19** |
| MixHop | 80.08 ± 1.43 | 79.39 ± 0.77 | 78.81 ± 0.66 | 79.50 ± 0.73 | **80.32 ± 0.71** | 70.88 ± 0.94 | 70.54 ± 1.27 | 70.15 ± 0.68 | 70.81 ± 1.16 | **71.50 ± 0.49** |
| Datasets | Actor (Heterophily) | | | | | Wisconsin (Heterophily) | | | | |
| Models | Grid | Random | Coarse | C2F | GNN-Diff | Grid | Random | Coarse | C2F | GNN-Diff |
| MLP | 37.36 ± 0.71 | 37.81 ± 0.71 | 37.22 ± 0.77 | **37.91 ± 0.79** | 37.89 ± 0.33 | 78.63 ± 1.72 | 78.43 ± 2.26 | 78.43 ± 4.13 | 79.61 ± 3.09 | **80.39 ± 1.07** |
| GCN | 30.94 ± 0.46 | 30.43 ± 0.47 | 30.79 ± 0.46 | 30.79 ± 0.46 | **31.24 ± 0.26** | 59.80 ± 2.96 | 59.41 ± 2.62 | 59.02 ± 1.72 | 60.39 ± 3.40 | **61.17 ± 0.78** |
| SAGE | 35.54 ± 0.84 | 35.53 ± 0.84 | 35.35 ± 0.83 | **36.70 ± 0.81** | 36.11 ± 0.09 | 75.68 ± 4.26 | 74.90 ± 4.60 | 73.73 ± 4.15 | 73.73 ± 5.48 | **76.47 ± 2.32** |
| APPNP | 35.32 ± 0.53 | 35.09 ± 0.53 | 35.11 ± 0.75 | 35.53 ± 0.76 | **35.92 ± 0.21** | 80.98 ± 3.59 | 79.61 ± 3.60 | 79.60 ± 3.60 | 80.78 ± 3.89 | **81.16 ± 1.53** |
| GAT | 29.84 ± 0.61 | 29.79 ± 0.86 | 29.61 ± 0.73 | 29.87 ± 0.74 | **29.88 ± 0.23** | **53.92 ± 3.73** | 51.96 ± 4.26 | 52.35 ± 2.62 | 51.57 ± 4.72 | 52.94 ± 0.05 |
| ChebNet | 37.29 ± 0.75 | 36.48 ± 0.72 | 36.23 ± 0.53 | 37.39 ± 0.69 | **37.49 ± 0.16** | 80.00 ± 4.32 | 79.61 ± 2.95 | 79.61 ± 2.48 | 79.80 ± 4.03 | **80.59 ± 2.83** |
| H2GCN | 34.05 ± 0.76 | 34.05 ± 0.76 | 34.01 ± 0.59 | 33.63 ± 0.65 | **34.16 ± 0.21** | 83.14 ± 2.65 | 82.55 ± 3.39 | 80.98 ± 3.93 | 82.55 ± 3.39 | **83.72 ± 0.90** |
| SGC | 30.32 ± 0.39 | 30.32 ± 0.39 | 29.13 ± 0.32 | 28.87 ± 0.42 | **30.38 ± 0.17** | 57.25 ± 4.41 | 58.24 ± 3.34 | 57.84 ± 4.06 | 57.25 ± 4.41 | **60.32 ± 1.19** |
| GPRGNN | 36.32 ± 0.43 | 36.56 ± 1.21 | 36.26 ± 0.61 | 36.68 ± 0.57 | **36.84 ± 0.41** | 80.20 ± 3.63 | 79.41 ± 3.83 | 79.41 ± 3.49 | 80.78 ± 4.41 | **81.53 ± 0.96** |
| MixHop | 37.76 ± 1.00 | 37.57 ± 0.92 | 37.06 ± 0.59 | 37.60 ± 0.60 | **38.15 ± 0.38** | 77.45 ± 3.10 | 79.41 ± 2.31 | 79.61 ± 2.95 | 79.80 ± 3.07 | **80.39 ± 0.07** |

Table 2: Results of node classification on large graphs. We report the average test accuracy (%) and the standard deviation of models selected by validation over 10 (`Flickr` and `OGB-arXiv`) or 3 `Reddit` and `OGB-Products`) training or generation runs. The best results are marked in **bold**.

| Datasets | Flickr | | | | | Reddit | | | | |
|---|---|---|---|---|---|---|---|---|---|---|
| Models | Grid | Random | Coarse | C2F | GNN-Diff | Grid | Random | Coarse | C2F | GNN-Diff |
| GCN | 52.54 ± 0.89 | 52.65 ± 0.99 | 52.64 ± 0.63 | 52.81 ± 0.76 | **52.84 ± 0.04** | 91.01 ± 0.09 | 91.07 ± 0.05 | 91.12 ± 0.07 | 91.12 ± 0.07 | **91.15 ± 0.04** |
| SAGE | 53.60 ± 0.39 | 53.53 ± 0.39 | 53.48 ± 0.47 | 53.63 ± 0.92 | **53.70 ± 0.02** | 93.26 ± 0.02 | 93.19 ± 0.06 | 93.28 ± 0.03 | 93.34 ± 0.06 | **93.45 ± 0.03** |
| APPNP | **52.37 ± 0.61** | 51.78 ± 0.64 | 51.92 ± 0.46 | 51.99 ± 0.21 | 52.23 ± 0.03 | 87.94 ± 0.07 | 86.44 ± 0.18 | 86.40 ± 0.12 | 87.93 ± 0.09 | **88.07 ± 0.05** |

| Datasets | OGB-arXiv | | | | | OGB-Products | | | | |
|---|---|---|---|---|---|---|---|---|---|---|
| Models | Grid | Random | Coarse | C2F | GNN-Diff | Grid | Random | Coarse | C2F | GNN-Diff |
| GCN | 69.22 ± 0.09 | 68.97 ± 0.05 | 68.53 ± 0.2 | 69.25 ± 0.15 | **69.38 ± 0.04** | 73.21 ± 0.19 | 73.29 ± 0.21 | 73.04 ± 0.23 | 73.94 ± 0.22 | **74.03 ± 0.16** |
| SAGE | 69.91 ± 0.38 | 69.85 ± 0.28 | 69.64 ± 0.17 | 70.03 ± 0.26 | **70.34 ± 0.07** | 75.68 ± 0.37 | 75.9 ± 0.33 | 75.6 ± 0.34 | 76.01 ± 0.42 | **76.52 ± 0.14** |
| APPNP | 55.05 ± 2.37 | 54.64 ± 4.05 | 54.58 ± 1.28 | 54.89 ± 1.12 | **55.09 ± 1.02** | **58.89 ± 0.21** | 56.09 ± 0.24 | 53.38 ± 0.36 | 54.22 ± 0.27 | 53.59 ± 0.18 |

Table 3: Results of node classification on long-range graphs. We report the average test F1-score and the standard deviation of models selected by validation over 3 training or generation runs. We show $100\times$ the original values for presentation purpose. The best results are marked in **bold**.

| Datasets | PascalVOC-SP | | | | | COCO-SP | | | | |
|---|---|---|---|---|---|---|---|---|---|---|
| Models | Grid | Random | Coarse | C2F | GNN-Diff | Grid | Random | Coarse | C2F | GNN-Diff |
| MLP | 11.82 ± 0.16 | 11.79 ± 0.12 | 11.77 ± 0.14 | 11.89 ± 0.27 | **11.97 ± 0.04** | 3.15 ± 0.31 | 3.13 ± 0.16 | 3.05 ± 0.26 | 3.17 ± 0.42 | **3.20 ± 0.05** |
| GCN | 23.31 ± 0.14 | 23.20 ± 0.17 | 23.18 ± 0.11 | 23.33 ± 0.26 | **23.52 ± 0.08** | 7.94 ± 0.21 | 7.78 ± 0.26 | 7.92 ± 0.18 | 7.90 ± 0.31 | **8.07 ± 0.03** |
| SAGE | 27.98 ± 0.23 | 28.08 ± 0.32 | 27.42 ± 0.26 | 27.98 ± 0.23 | **28.24 ± 0.06** | 8.65 ± 0.61 | 9.19 ± 0.34 | 8.99 ± 0.64 | 8.65 ± 0.61 | **9.28 ± 0.08** |
| APPNP | **17.24 ± 0.12** | 15.82 ± 0.11 | 15.25 ± 0.14 | **17.24 ± 0.12** | 15.57 ± 0.05 | 4.28 ± 0.07 | 4.27 ± 0.08 | 4.06 ± 0.15 | 4.27 ± 0.08 | **4.43 ± 0.02** |
| SGC | 21.65 ± 0.23 | 21.55 ± 0.21 | 21.05 ± 0.26 | 21.32 ± 0.14 | **21.80 ± 0.12** | 5.56 ± 0.39 | 5.82 ± 0.29 | 5.53 ± 0.40 | 5.70 ± 0.33 | **5.84 ± 0.11** |
| GPRGNN | **15.56 ± 0.19** | 15.21 ± 0.10 | 15.03 ± 0.09 | 14.91 ± 0.12 | 15.26 ± 0.09 | **4.40 ± 0.07** | 4.33 ± 0.12 | 4.27 ± 0.08 | 4.39 ± 0.08 | 4.34 ± 0.07 |
| MixHop | 21.79 ± 0.18 | 22.37 ± 0.12 | 22.60 ± 0.16 | 22.60 ± 0.16 | **22.75 ± 0.03** | **7.06 ± 0.19** | 7.01 ± 0.15 | 6.55 ± 0.25 | 6.87 ± 0.21 | 6.74 ± 0.03 |

Table 4: Results of link existence prediction. We report the average test accuracy (%) and the standard deviation of models selected by validation over 10 training or generation runs. The best results are marked in **bold**.

| Datasets | Cora | | | | | Citeseer | | | | |
|---|---|---|---|---|---|---|---|---|---|---|
| Models | Grid | Random | Coarse | C2F | GNN-Diff | Grid | Random | Coarse | C2F | GNN-Diff |
| MLP | 75.04 ± 1.12 | 74.61 ± 1.72 | 74.05 ± 1.25 | 74.79 ± 0.97 | **75.42 ± 0.19** | 78.26 ± 0.80 | 78.23 ± 1.23 | 76.79 ± 1.28 | 78.25 ± 1.79 | **78.87 ± 0.07** |
| GCN | 76.92 ± 0.83 | 76.73 ± 0.80 | 76.29 ± 0.88 | 76.47 ± 0.43 | **77.04 ± 0.12** | 77.90 ± 0.90 | 75.93 ± 0.72 | 76.90 ± 0.92 | 77.25 ± 0.92 | **77.94 ± 0.14** |
| SAGE | 75.09 ± 0.49 | 75.22 ± 1.14 | 74.97 ± 0.43 | 74.11 ± 2.24 | **76.33 ± 0.31** | 76.25 ± 1.15 | 76.37 ± 1.32 | 75.84 ± 1.91 | 76.43 ± 0.85 | **76.51 ± 0.08** |
| APPNP | 75.95 ± 0.93 | 75.80 ± 1.07 | 75.64 ± 0.91 | 75.95 ± 0.93 | **76.94 ± 0.15** | 75.98 ± 1.06 | 75.73 ± 0.81 | 75.68 ± 0.72 | 75.92 ± 0.85 | **76.15 ± 0.15** |
| ChebNet | **74.96 ± 0.79** | 74.53 ± 0.70 | 74.17 ± 0.48 | 74.31 ± 0.96 | **74.96 ± 0.15** | 78.31 ± 1.88 | 77.36 ± 2.77 | 76.76 ± 1.38 | 76.82 ± 0.82 | **78.35 ± 0.33** |

| Datasets | Chameleon | | | | | Squirrel | | | | |
|---|---|---|---|---|---|---|---|---|---|---|
| Models | Grid | Random | Coarse | C2F | GNN-Diff | Grid | Random | Coarse | C2F | GNN-Diff |
| MLP | **76.88 ± 0.39** | 76.56 ± 0.46 | 75.27 ± 0.66 | 76.61 ± 0.50 | 76.23 ± 0.10 | 73.24 ± 0.16 | 73.13 ± 0.07 | 73.13 ± 0.07 | 73.28 ± 0.22 | **73.32 ± 0.04** |
| GCN | **78.71 ± 0.54** | 78.50 ± 0.38 | 77.69 ± 0.43 | 77.91 ± 0.34 | 78.26 ± 0.02 | 74.72 ± 0.12 | 74.52 ± 0.45 | 74.61 ± 0.11 | 74.62 ± 0.17 | **74.73 ± 0.05** |
| SAGE | 82.06 ± 0.41 | 81.89 ± 0.35 | 81.88 ± 0.40 | 82.02 ± 0.51 | **82.08 ± 0.19** | **75.34 ± 0.32** | 74.49 ± 0.51 | 74.09 ± 0.26 | 74.37 ± 0.27 | 74.66 ± 0.02 |
| APPNP | **78.87 ± 0.75** | 77.67 ± 0.34 | 77.66 ± 0.29 | 77.74 ± 0.65 | 77.95 ± 0.04 | 73.61 ± 0.22 | 73.54 ± 0.25 | 73.51 ± 0.17 | 73.61 ± 0.22 | **73.76 ± 0.09** |
| ChebNet | 81.00 ± 0.54 | 77.52 ± 0.46 | 77.25 ± 0.44 | 77.25 ± 0.44 | **81.30 ± 0.18** | 75.48 ± 0.34 | 75.31 ± 0.32 | 73.56 ± 0.62 | 75.32 ± 0.43 | **75.54 ± 0.03** |

## 5.2 Main Experiment Results

**Results of basic node classification.** We show results of node classification on `Cora`, `Citeseer`, `Actor`, and `Wisconsin` in Table 1. More node classification results on homophilic and heterophilic graphs can be found in Appendix I. In general, models generated by GNN-Diff achieve the best average test accuracy for 36 out of 40 experiments in Table 1. For the rest of 4 experiments, GNN-Diff produces accuracy that still sufficiently matches the grid search performance. Both GNN-Diff and C2F show clear improvement on the coarse search results, while GNN-Diff outperforms C2F for most target models and datasets. Moreover, it is apparent that models generated by GNN-Diff exhibit more stable performance in terms of standard deviation.

**Results of node classification on large graphs.** The challenge of large graphs is often associated with the large number of nodes, which makes the computation of graph convolution considerably slow. In addition, GNNs often need a comparably larger hidden size (e.g., 256) to achieve good performance on large graphs (Hu et al., 2020a), which increases the volume of parameters to learn. The experiment results in Table 2 show that GNN-Diff can produce high-performing GNNs on large

graphs by generating only the last layer parameters efficiently. In consideration of time and memory efficiency, our experiments are conducted with the clustering training (Chiang et al., 2019). We also discuss the potential of combining GNN-Diff with other state-of-the-art large-scale training algorithms in Appendix I.7.

**Results of node classification on long-range graphs.** In long-range graphs, nodes need to exchange information over long distances (i.e., information is passed through many edges to form useful graph representations), which usually relies on very deep GNNs with many graph convolutional layers (Dwivedi et al., 2022). Partial generation is even more helpful in this case because a well-informed graph representation is formed just before the last layer, making it an important mapping to final classification. We observe in Table 3 that GNN-Diff generates top-performing GNNs for most long-range graph tasks as expected.

**Results of link prediction.** To validate GNN-Diff on various graph tasks, we conduct link existence prediction following the experiment design in (Kipf & Welling, 2016). We use accuracy as the metric instead of AUC-ROC because we adopt equal-proportion sampling for positive and negative edges. The results are presented in Table 4. Models generated by GNN-Diff show outstanding performance in `Cora`, `Citeseer`, and `Squirrel`. On `Chameleon`, however, random search shows more promising results, though GNN-Diff generally better enhances the original performance from coarse search compared to C2F.

## 5.3 FURTHER DISCUSSIONS

**Visualizations**

We use visualizations of node classification with GCN on `Citeseer` as examples to answer the following three questions. Please note that "input samples" refers to the parameter samples collected for GNN-Diff training.

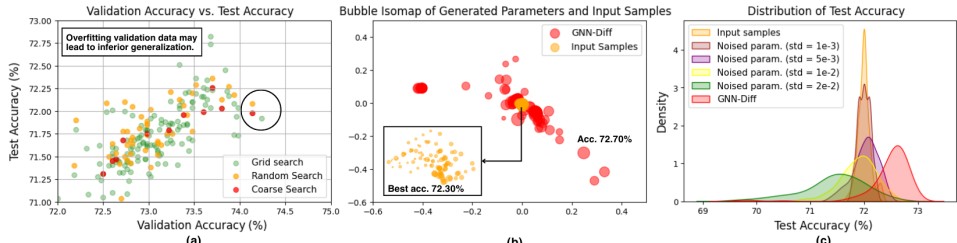

Figure 4: (a) Scatter plot between validation and test accuracy for grid, random, and coarse search. The black circle indicates the final models selected by validation. (b) Visualization of generated and input sample parameters with isomap. Each bubble represents one parameter sample, and the bubble size shows the corresponding test accuracy. (c) Kernel density estimation plot of test accuracy distributions corresponding to input samples, noised sample parameters, and GNN-Diff parameters.

**Q1. Why are some random or coarse search results even better than grid search?** We observe that some random or coarse search results may be even better than the comprehensive grid search. For example, with GCN node classification on `Citeseer`, results from coarse, random, and grid search are 71.97%, 72.04%, and 71.92%, respectively. This is different from our previous observation in Figure 1. In Figure 4 (a), we note that models that achieve higher validation accuracy may not perform better on the test set. We recognize this phenomenon as "overfitting on validation set" with overly extensive grid search. Random search with a smaller search space may solve the problem. But how do we know the appropriate search space size? In addition, more than 80% of experiment results show that grid search can find better generalized models than random search. Thus GNN-Diff, which learns from samples with sub-optimal validation performance, is considered as a better solution.

**Q2. How does GNN-Diff generate better performance based on the input samples?** In short, GNN-Diff approximates the population of input samples and explores into better-performing regions. In Figure 4 (b), we visualize the input parameters (orange) and 100 parameters generated by GNN-Diff (red) via isometric mapping dimensionality reduction (Tenenbaum et al., 2000). Additionally, we plot each model point as bubbles with bubble size representing the corresponding test accuracy. We

observe that parameters generated by GNN-Diff spread around the input sample distribution. Notably, parameters in the extended distribution may lead to better accuracy than the best-performing input sample (72.70% vs. 72.30%). This may verify our assumption that better-performing parameters exist in the population of sample parameters from a sub-optimal configuration, and can be found by GNN-Diff.

**Q3. Can we find better-performing models by simply adding random noises to input samples?**
There is a slight possibility, but GNN-Diff clearly shows more promising outcomes. To answer this question, we conduct an experiment in which we add random Gaussian noises to the input samples of GNN-Diff. We set the mean of noises as 0 and tune the standard deviation (std) in [1$e$-3, 5$e$-3, 1$e$-2, 2$e$-2]. All distributions are plotted based on 100 results. The model performance with lightly noised parameters (std = 1$e$-3) almost follows the sample distribution, while high-level noises (std = 2$e$-2) significantly diminish the model accuracy. We observe that when noise std = 5$e$-3, the corresponding accuracy distribution presents slightly better results than the original samples. Nevertheless, parameters from GNN-Diff consistently outperform the noised ones, indicating that GNN-Diff enhances model performance beyond random guessing.

**Full Generation vs. Partial Generation**

We found that GNN-Diff significantly enhances GNNs by generating only the last learnable layer of the target model. To confirm the effectiveness of this approach, we compared GNN-Diff with three different generation and sampling strategies: (a) partial generation as in the main experiments; (b) full generation (best checkpoint) with samples collected by training the best validation checkpoint from 190 epochs with another 10 epochs and a very small learning rate for 10 runs; (c) full generation (all checkpoints) with samples collected by saving all checkpoints over 200 epochs and discarding the first 10 that have not yet converged.

In Figure 5, we show the distribution of the GNN-Diff parameters and the input parameters and their test accuracy. The test accuracy distribution of GNN-Diff is based on the results of 30 generation runs of 100 samples. We only show the test accuracy corresponding to the best validation sample in each run. Overall, GNN-Diff effectively captures and expands the sample distribution across all three strategies. Moreover, the generated samples usually yield better test performance than the input samples. Partial generation and full generation (best checkpoint) produce similar results, while full generation (all checkpoints) shows noticeably inferior performance. This difference may be caused by the low-quality parameters in the input samples, as the test accuracy from the input samples collected with the all-checkpoint strategy exhibits a much wider range of lower values (see Figure 5 (c)).

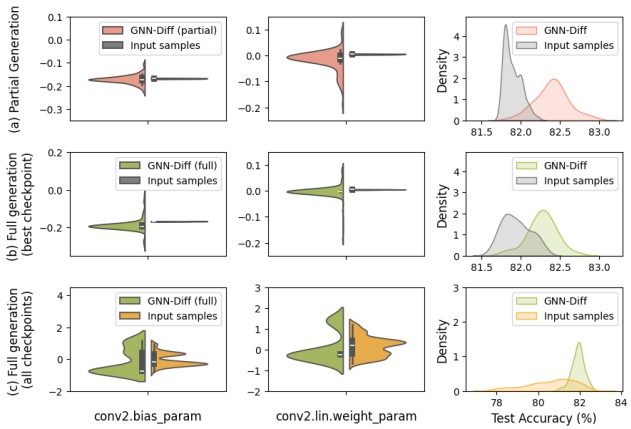

Figure 5: GNN-Diff for GCN node classification on `Cora` with three generation and sampling strategies. Each row contains visualisations of (1) generated and input sample distributions of the first parameter in last layer bias; (2) generated and input sample distributions of the first parameter in last layer weights; (3) test accuracy of generated and input sample parameters.

**Ablation Study on the Graph Condition**

An ablation study is conducted to evaluate GNN parameter generation with and without the graph condition (more analyses related to the GFE and graph condition are in Appendix I.5). Here we compare GNN-Diff with p-diff (Wang et al., 2024), an unconditional diffusion model for parameter generation. GNN-Diff adopts nearly identical PAE and $\epsilon_\theta$ as p-diff, minimizing the impact of model architecture differences. We apply the same training and hyperparameter settings for both methods to ensure that the only difference is the graph condition. The results are exhibited in Table 5. GNN-Diff outperforms p-diff in almost all experiments, and generally presents more stable generation outcomes

with lower standard deviation. Therefore, we conclude that the graph condition is indeed helpful with GNN parameter generation by providing insightful guidance based on graph and data information.

Table 5: Comparison between GNN parameter generation with and without the graph condition (GNN-Diff vs. p-diff). We consider node classification on 4 types of graphs, `Cora` (homophily), `Actor` (heterophily), `Flickr` (large), and `PascalVOC-SP` (long-range), and two target models, GCN and SAGE.

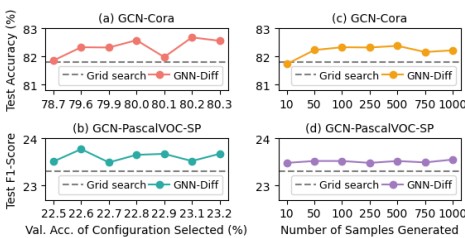

Figure 6: Sensitivity analysis. How average test results varies with coarse search results (left column) and number of samples generated (right column).

| Models | GCN | | SAGE | |
|---|---|---|---|---|
| Methods | GNN-Diff | p-diff | GNN-Diff | p-diff |
| **Cora** | **82.33 ± 0.17** | 81.96 ± 0.31 | **80.60 ± 0.05** | 80.19 ± 0.60 |
| **Actor** | **31.24 ± 0.26** | 30.96 ± 0.30 | 36.11 ± 0.09 | **36.25 ± 0.31** |
| **Flickr** | **52.84 ± 0.04** | 52.70 ± 0.11 | **53.70 ± 0.02** | 53.55 ± 0.23 |
| **PascalVOC-SP** | **23.52 ± 0.08** | 23.49 ± 0.09 | **28.24 ± 0.06** | 28.12 ± 0.02 |

**Sensitivity Analyses**

**(1) Sensitivity analysis on coarse search results.** We would like to know whether a coarse search with 10% of full search space is able to find a sufficiently good configuration for GNN-Diff to generate high-performing parameters. So, we repeat the coarse search for GCN node classification on `Cora` and `PascalVOC-SP` without fixed random seed 7 times and obtain 7 configurations selected for parameter collection. Then, we train and generate with GNN-Diff based on these configurations. The results show that GNN-Diff consistently produces better test outcomes than grid search (see Figure 6 (a) and (b)), which confirms the feasibility of adopting the 10% coarse search space.

**(2) Sensitivity analysis on the number of samples generated by GNN-Diff.** Since overfitting the validation data may lead to inferior results on unseen data, we wonder if generating too many samples from GNN-Diff will encounter this issue. Hence, we set the number of samples generated by GNN-Diff from 10 to 1000. In general, GNN-Diff's performance on unseen data is not very sensitive to the number of samples, though generating 750 to 1000 samples indeed diminishes the test accuracy on `Cora` (see Figure 6 (c) and (d)). In consideration of sampling efficiency, we suggest that 100 samples are sufficient for GNN-Diff to produce good performance in a reasonable sampling time.

**Time Efficiency**

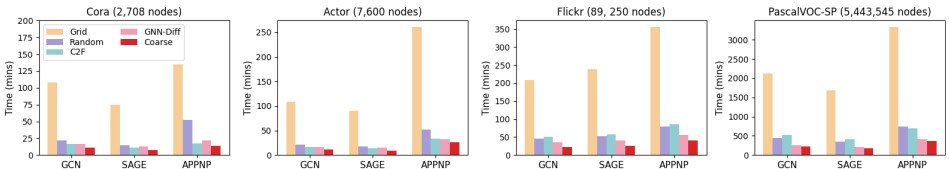

Figure 7: Time costs of grid search, random search, coarse search, C2F search, and GNN-Diff.

Finally, we compare the time costs of GNN-Diff and other search methods in Figure 7. The coarse search is the most time-consuming component in GNN-Diff (see more details in Appendix I.6). GNN-Diff shows a clear time advantage, though it may not be faster than C2F on small graphs such as `Cora` and `Actor`. This is acceptable, as GNN-Diff often yields better-performing models than C2F and is more efficient on larger graphs such as `Flickr` and `PascalVOC-SP`, which offers greater practical value, especially for complicated tasks with large and long-range graphs.

## 6  CONCLUSION

In this paper, we proposed a graph-conditioned latent diffusion framework, GNN-Diff, to generate top-performing GNN parameters by learning from checkpoints saved with a sub-optimal configuration selected by a light-tuning coarse search. We validate with 166 empirical experiments that our method is an efficient alternative to costly search methods and is able to generate better prediction outcomes than the comprehensive grid search on unseen data. Future works may involve combining GNN-Diff with efficient large-scale training algorithms, and more details will be discussed in I.7.

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

## Appendices Contents

# A RELATED WORKS

**Network Parameter Generation**

Hypernetwork (Ha et al., 2017; Stanley et al., 2009) is an early concept of deep learning technique for network generation and now serves as a general framework that encompasses many existing methods. A hypernetwork is a neural network that learns how to predict the parameters of another neural network (target network). The input of a hypernetwork may be as simple as parameter positioning embeddings (Stanley et al., 2009), or more informative, such as encodings of data, labels, and tasks (Deutsch, 2018; Ha et al., 2017; Ratzlaff & Fuxin, 2019; Zhang et al., 2019; Zhmoginov et al., 2022).

With the emergence of generative modelling, studies on distributions of network parameters have gained their popularity in relevant research. A pioneering work of (Unterthiner et al., 2020) showed that simply knowing the statistics of parameter distributions enabled us to predict model accuracy without accessing the data. Enlightened by this finding, two following works (Schürholt et al., 2022; 2021) studied the reconstruction and sampling of network parameters from their population distribution. The similar idea was also investigated by (Peebles et al., 2022), who proposed that network parameters corresponding to a specific metric value of some specific data and tasks could be generated with a transformer-diffusion model trained on a tremendous amount of model checkpoints collected from past training on a group of data and tasks. This method, however, relies highly on the amount and quality of model checkpoints fed to the generative model. A more network- and data-specific approach, p-diff (Wang et al., 2024), alternatively collects checkpoints from the training process of the target network and uses an unconditional latent diffusion model to generate better-performing parameters. Another recent work, D2NWG (Soro et al., 2024), adopts dataset-conditioned latent diffusion to generate parameters for target networks on unseen datasets. Diffusion-based network generation has also been widely applied to other applications, such as meta learning (Nava et al., 2022; Zhang et al., 2024) and producing implicit neural-filed for 3D and 4D synthesis (Erkoç et al., 2023).

**GNN Training**

Previous works on GNN training generally focus on two topics: GNN pre-training and GNN architecture search. GNN pre-training involves using node, edge, or graph level tasks to find appropriate initialization of GNN parameters before fine-tuning with ground truth labels (Hu et al., 2020b;c; Lu et al., 2021). This approach still inevitably relies on hyperparameter tuning to boost model performance. GNN architecture search aims to find appropriate GNN architectures given data and tasks (Gao et al., 2020; You et al., 2020). However, it was shown in (Shchur et al., 2018) that under a comprehensive hyperparameter tuning and a proper training process, simple GNN architectures such as GCN may even outperform sophisticated ones.

**Advanced Methods for Hyperparameter Tuning**

There emerge many recent works on parameter-free optimization, such as (Defazio & Mishchenko, 2023; Ivgi et al., 2023; Mishchenko & Defazio, 2024) that automatically set step size based on problem characteristics. However, most if not all methods require nontrivial adaptation of existing optimizers and show theoretical guarantees only for convex functions. In addition, the parameter-free optimization is only designed for tuning training-related hyperparameters such as the learning rate. This means it is less suitable for GNN hyperparameter tuning, which usually involves model-related hyperparameters such as the teleport probability of APPNP. Bayesian optimization (Snoek et al., 2012; 2015), on the other hand, is a more advanced approach to tuning hyperparameters in a more informed way than grid and random search. Bayesian optimization usually adopts a probabilistic surrogate model to approximate the loss function and then uses an acquisition function to sample hyperparameters for further search. Additionally, Hyperband (Li et al., 2018), a method that is considered more efficient than Bayesian optimization, optimizes the search space by allocating resources (e.g., training time or dataset size) to more promising configurations and pruning those that perform poorly. In our experiments, we consider the coarse-to-fine (C2F) search (Moshkelgosha et al., 2017; Payrosangari et al., 2020), a variant of random search and Hyperband that narrows the search space based on the result of a coarse search.

## B  PSEUDO CODE OF GNN-DIFF

---

**Algorithm 1** Algorithm of GNN-Diff Parameter Generation

---

**Input:** Graph condition $\mathbf{c}_{\mathcal{G}}$ from Algorithm 2; the learned P-Decoder($\cdot$) from Algorithm 3; the learned denoising network $\epsilon_{\theta}$ from Algorithm 4.
**Output:** Generated parameters $\hat{\mathbf{w}}_0$ that can be returned to target GNN for prediction.
  1: Run Algorithm 5 to sample latent parameter $\hat{\mathbf{z}}_0$
  2: Reconstruct parameters with the learned PAE decoder: $\hat{\mathbf{w}}_0 = \text{P-Decoder}(\hat{\mathbf{z}}_0)$

---

**Algorithm 2** Training Algorithm of GFE (Basic Node Classification Example)

---

**Input:** Graph signals $\mathbf{X}$; graph structure $\mathbf{A}$; graph label $\mathbf{Y}$; graph training mask $M_{train}$; number of training epochs $E_{\text{GFE}}$.
**Output:** Graph condition $\mathbf{c}_{\mathcal{G}}$.
  1: **for** $i = 1, 2, ..., E_{GFE}$ **do**
  2:    Compute G-Encoder: $\eta = \text{Concat}\left(\mathbf{A}^2\mathbf{X}\mathbf{W}_1, \mathbf{A}\mathbf{X}\mathbf{W}_2, \mathbf{X}\mathbf{W}_3\right)\mathbf{W}_4$
  3:    Compute G-Decoder: $\hat{\mathbf{Y}} = \text{Softmax}(\eta\mathbf{W}_5)$
  4:    Take gradient step and update $\mathbf{W} = \{\mathbf{W}_1, \mathbf{W}_2, \mathbf{W}_3, \mathbf{W}_4, \mathbf{W}_5\}$:

$$\nabla_{\mathbf{W}}\text{CrossEntropy}(\mathbf{Y} \odot M_{train}, \hat{\mathbf{Y}} \odot M_{train})$$

  5: **end for**
  6: Compute graph representation $\eta = \text{Concat}\left(\mathbf{A}^2\mathbf{X}\mathbf{W}_1, \mathbf{A}\mathbf{X}\mathbf{W}_2, \mathbf{X}\mathbf{W}_3\right)\mathbf{W}_4$
  7: Conduct mean pooling to obtain graph condition $\mathbf{c}_{\mathcal{G}} = \frac{1}{N}\sum_{i=1}^{N}\eta(i)$.

---

**Algorithm 3** Training Algorithm of PAE

---

**Input:** Vectorized parameters collected based on coarse search $\mathbf{w}_0 \sim q_{\mathbf{w}}(\mathbf{w}_0)$; number of training epochs $E_{\text{PAE}}$.
**Output:** Latent parameter representation $\mathbf{z}_0 \sim q_{\mathbf{z}}(\mathbf{z}_0)$; the learned decoder P-Decoder($\cdot$).
  1: **for** $i = 1, 2, ..., E_{\text{PAE}}$ **do**
  2:    Compute latent parameter representation $\widetilde{\mathbf{z}}_0 = \text{P-Encoder}(\mathbf{w}_0)$
  3:    Compute PAE decoder for reconstruction $\widetilde{\mathbf{w}}_0 = \text{P-Decoder}(\widetilde{\mathbf{z}}_0)$
  4:    Take gradient step and update PAE parameters with the loss function $\mathcal{L}_{\text{PAE}} = \|\mathbf{w}_0 - \widetilde{\mathbf{w}}_0\|^2$
  5: **end for**
  6: Use PAE encoder to produce latent parameter representation $\mathbf{z}_0 = \text{P-Encoder}(\mathbf{w}_0)$.

---

**Algorithm 4** Training Algorithm of $\epsilon_{\theta}$ in G-LDM

---

**Input:** Latent parameter representation $\mathbf{z}_0 \sim q_{\mathbf{z}}(\mathbf{z}_0)$; graph signals $\mathbf{X}$; graph structure $\mathbf{A}$; number of diffusion steps $T$; noise schedule $\beta = \{\beta_1, \beta_2, ..., \beta_T\}$; number of training epochs $E_{\text{G-LDM}}$.
**Output:** The learned denoising network $\epsilon_{\theta}$.
  1: Sample $\mathbf{z}_0 \sim q_{\mathbf{z}}(\mathbf{z}_0)$
  2: **for** $i = 1, 2, ..., E_{\text{G-LDM}}$ **do**
  3:    $t \sim \text{Uniform}(1, T), \epsilon \sim \mathcal{N}(\mathbf{0}, \mathbf{I})$
  4:    Take gradient step and update $\theta$:

$$\nabla_{\theta}\mathbb{E}_{t,\mathbf{z}_0 \sim q_{\mathbf{z}}(\mathbf{z}_0),\epsilon \sim \mathcal{N}(\mathbf{0},\mathbf{I})}\left\|\epsilon - \epsilon_{\theta}\left(\sqrt{\widetilde{\alpha}_t}\mathbf{z}_0 + \sqrt{1 - \widetilde{\alpha}_t}\epsilon, t, \mathbf{c}_{\mathcal{G}}\right)\right\|^2$$

  5: **end for**

---

---

**Algorithm 5** Inference Algorithm of G-LDM

---

**Input:** Number of diffusion steps $T$; noise schedule $\beta = \{\beta_1, \beta_2, ..., \beta_T\}$; diffusion sampling variance hyperparameter $\sigma = \{\sigma_1, \sigma_2, ..., \sigma_T\}$.
**Output:** Generated latent parameters $\hat{\mathbf{z}}_0$.
 1: Randomly generate white noises $\hat{\mathbf{z}}_T \sim \mathcal{N}(\mathbf{0}, \mathbf{I})$
 2: **for** $t = T, T-1, ..., 1$ **do**
 3:     $\mathbf{e} = 0$ if $t = 1$ else $\mathbf{e} \sim \mathcal{N}(\mathbf{0}, \mathbf{I})$
 4:     Compute and update

$$\hat{\mathbf{z}}_{t-1} = \frac{1}{\sqrt{1 - \beta_t}} \left( \hat{\mathbf{z}}_t - \frac{\beta_t}{\sqrt{1 - \widetilde{\alpha}_t}} \epsilon_\theta(\hat{\mathbf{z}}_t, t, \mathbf{c}_{\mathcal{G}}) \right) + \sigma_t \mathbf{e}$$

 5: **end for**

---

## C  DETAILS OF GNN-DIFF

### C.1  GFE ARCHITECTURES FOR VARIOUS TASKS

We provide illustrations of the GFE architectures for various tasks in Figure 8. In GFE encoders, we generally adopt graph convolutional layers to process graph data and structural information. The decoders, on the other hand, is a single linear layer (MLP1), thus the graph information is entirely preserved in the output of the encoders, which will later be further processed and used as the graph condition in G-LDM. The details of GFE for each task are as follows. Please note that for simplicity, we show encoders without activation functions, but activation functions can be included in implementation.

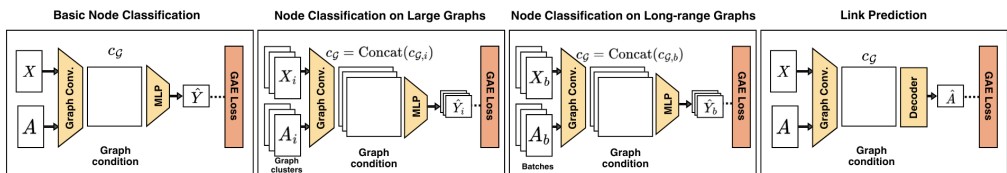

Figure 8: GFE architectures for node classification on small, large, and long-range graphs, and link prediction. The graph condition is designed to incorporate both data and graph structural information and guided by specific tasks and the corresponding loss functions.

**Basic node classification.** Details can be found in Subsection 4.2.

**Node classification on large graphs.** The input to the GFE is a set of graph clusters, each with $N_i$ nodes for $i = 1, 2, ..., I$. Details of how the graph clusters are obtained from the original large graph with $N$ nodes are discussed in G.2. The feature matrix and adjacency matrix of each cluster are denoted as $\mathbf{X}_i$ and $\mathbf{A}_i$, respectively. We adopt the same encoder architecture as for basic node classification, because the homophilic and heterophilic problem may also exist in large graph node classification. This may or may not be true for the datasets in our experiments, but we design in this way for future extension to other large graphs. Hence, for each cluster, the encoder is applied following Equation 2 as

$$\eta_i = \text{G-Encoder}(\mathbf{X}_i, \mathbf{A}_i) = \text{Concat}\left(\mathbf{A}_i^2 \mathbf{X}_i \mathbf{W}_1, \mathbf{A}_i \mathbf{X}_i \mathbf{W}_2, \mathbf{X}_i \mathbf{W}_3\right) \mathbf{W}_4 \in \mathbb{R}^{N_i \times D_p}. \quad (4)$$

The decoder and loss function are the same as the GFE for basic node classification as well. To form the graph condition, we shall first concatenate the outputs of the encoder in the node dimension as

$$\eta = \text{Concat}(\eta_1, \eta_2, ..., \eta_I) \in \mathbb{R}^{N \times D_p}, \quad (5)$$

which will be eventually turned into the graph condition via the node mean-pooling.

**Node classification on long-range graphs.** The input of GFE is a set of batches, where each batch contains a set of small graphs. Suppose that we have $B$ batches and that a small graph in a batch has $N_{b,j}$ nodes for $b = 1, 2, ..., B$, and $j = 1, 2, ..., J_b$, where $J_b \in [J_1, J_2, ..., J_B]$ is the number of

small graphs in each batch. Then, we organize the small graphs in each batch into a combined batch graph with $N_b = N_{b,1} + N_{b,2} + \cdots + N_{b,J_b}$ nodes. Let the feature matrix and adjacency matrix of each batch graph formed by the small graphs in a batch be $\mathbf{X}_b$ and $\mathbf{A}_b$, respectively. The encoder will be applied to each batch graph as

$$\eta_b = \text{G-Encoder}(\mathbf{X}_b, \mathbf{A}_b) = \text{Deep-GCN}(\mathbf{X}_b, \mathbf{A}_b) \in \mathbb{R}^{N_b \times D_p}, \tag{6}$$

where Deep-GCN$(\cdot)$ is a GCN-based structure with 7 GCN convolutional layers and residual connections for the training purpose. We choose this deep GCN-based structure as the GFE encoder to better process long-term information among distant nodes, which is usually considered as important in long-range graphs. The decoder is as simple as a single linear layer. Moreover, the loss function is the weighted cross entropy discussed in the appendix G.3. Similar to large graphs, the graph condition is constructed by mean-pooling the concatenation of encoder outputs:

$$\eta = \text{Concat}(\eta_1, \eta_2, ..., \eta_B) \in \mathbb{R}^{N_{\text{all}} \times D_p}, \tag{7}$$

where $N_{\text{all}}$ is the total number of nodes of small graphs in all batches.

**Link prediction.** The GFE for link prediction is designed based on the implementation in (Kipf & Welling, 2016). Details of the link prediction task can be found in Appendix G.4. The input of GFE is the feature matrix $\mathbf{X}$ and the adjacency matrix $\mathbf{A}_{\text{train}}$ with only training edges. The encoder is designed as a 2-layer GCN:

$$\eta = \text{G-Encoder}(\mathbf{X}, \mathbf{A}_{\text{train}}) = \mathbf{A}_{\text{train}}(\mathbf{A}_{\text{train}}\mathbf{X}\mathbf{W}_1)\mathbf{W}_2 \in \mathbb{R}^{N \times D_p}, \tag{8}$$

where $\mathbf{W}_1$ and $\mathbf{W}_2$ are learnable linear operators. The decoder is again a single linear layer, but is followed by the pairwise node multiplication and embedding aggregation for link prediction. The loss function is binary cross entropy computed based on the training edge labels. Lastly, the graph condition can be constructed by mean-pooing the graph nodes.

## C.2 PAE AND G-LDM DENOISING NETWORK ARCHITECTURES

Illustrations of the PAE and the G-LDM denoising network $\epsilon_\theta$ are provided in Figure 9 and Figure 10, respectively. Taking the idea from (Wang et al., 2024), both architectures have `Conv1d`-based layers with LeakyReLU activation and instance normalization.

**PAE.** P-Encoder and P-Decoder are both composed of 4 basic blocks in Figure 9. Since the input is latent representation of vectorized parameters, the number of input channels of the first block in P-Encoder is 1. The number of output channels for the 4 blocks in P-Encoder is [6, 6, 6, 6]. Accordingly, the number of input channels of the first block in P-Decoder is also 6. The number of output channels of the 4 blocks in P-Decoder is [512, 512, 8, 1].

**G-LDM $\epsilon_\theta$.** The denoising network $\epsilon_\theta$ adopts an encoder-decoder structure with 8 basic blocks in Figure 10. The number of input channels and output channels, kernel size, and stride are the same as in (Wang et al., 2024). In Figure 10, we also mark the position where the graph condition $\mathbf{c}_\mathcal{G}$ is involved. We include the graph condition in each encoder and decoder block by adding it to the input of the block directly.

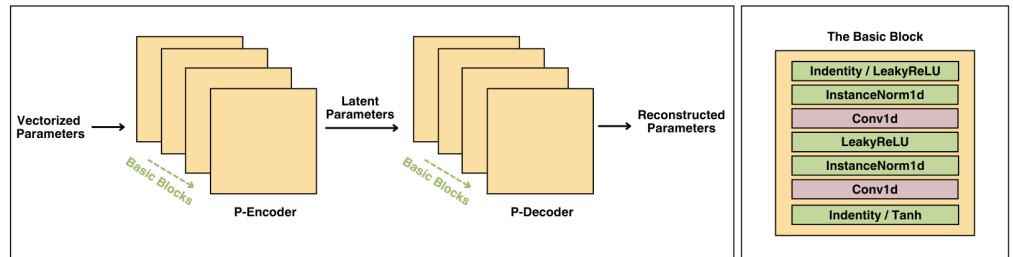

Figure 9: The PAE architecture. The PAE adopts an encoder-decoder structure with 4 encoder blocks and 4 decoder blocks. All blocks have the same structure as shown in the basic block. The dimensionality of the input parameters is first reduced by the P-Encoder, and then recovered by the P-Decoder.

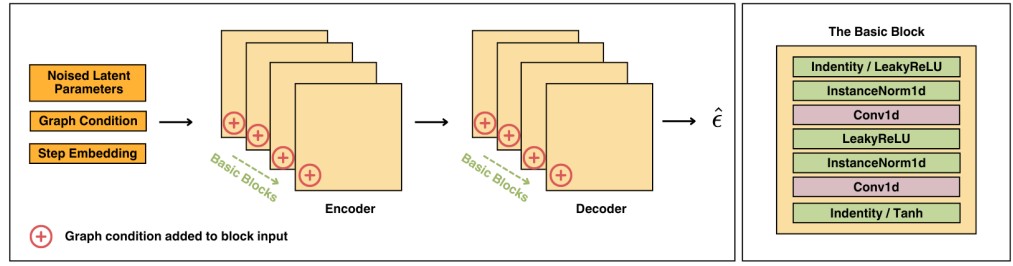

Figure 10: The $\epsilon_\theta$ architecture. Like PAE, $\epsilon_\theta$ adopts an encoder-decoder structure with 4 encoder blocks and 4 decoder blocks. We mark the position where the graph condition is involved by taking sum with the input of each block.

## C.3 GNN-DIFF SETTINGS

GFE is trained with the same graph data as target models. PAE is trained with vectorized parameter samples, while G-LDM learns from their latent representation. Batch size for PAE and G-LDM training is 50. All three modules are trained with the same AdamW optimizer (Loshchilov & Hutter, 2019) with learning rate 0.001 and weight decay 0.002. The concatenation dimension $D_{\mathcal{G}}$ is set the same as the latent parameter dimension $D_p$, which is decided by the Conv1d layer in PAE. Dropout is applied with dropout rate 0.1. For G-LDM, we set the number of diffusion steps $T = 1000$. We choose the linear beta schedule with $\beta_1 = 0.0001$ and $\beta_T = 0.02$.

**How to ensure high quality generation?** We observe in experiments that GNN-Diff does not rely on hyperparameter tuning to generate top-performing GNNs. The only factor that may influence the generative outcomes is the dimension of the latent parameters, which can be easily adjusted by setting the kernel size and stride of Conv1d layers in P-Encoder blocks. Additionally, the appropriate kernel size and stride can be easily decided based on the dimension of the original parameter samples (see Table 6). Therefore, we highlight that **GNN-Diff does not introduce additional hyperparameter tuning burdens**, and all tuning efforts are only associated with the tuning of target models in coarse search.

Table 6: Kernel size and stride of Conv1d layers in P-Encoder based on the input parameter dimension. The simple rule is: (1) the stride equals to the kernel size; (2) if the parameter dimension is larger than 1000, we increase kernel size and stride by 1 for every 5000 parameters.

| Vectorized parameter dimension | Kernel size | Stride |
|:---:|:---:|:---:|
| $\leq 99$ | 2 | 2 |
| 100 - 499 | 3 | 3 |
| 500 - 999 | 4 | 4 |
| 1000 - 4999 | 5 | 5 |
| 5000 - 9999 | 6 | 6 |
| 10000 - 14999 | 7 | 7 |
| 15000 - 19999 | 8 | 8 |
| 20000 - 24999 | 9 | 9 |
| ... | ... | ... |

## D BASELINE METHODS

We involve four baseline search methods to compare with GNN-Diff, including grid search, random search, coarse search, and C2F search. As aforementioned in Subsection 5.1, random search and coarse search adopt 20% and 10% of the full search space in grid search. In terms of C2F search, we first conduct the coarse search, and then fix the model-related hyperparameters and optimizer based on the coarse search result, and further tune other training-related hyperparameters. For example, if the hyperparameter configuration leading to the best validation accuracy is {optimizer: 'Adam', learning_rate: 0.01, weight_decay: 0.005, dropout: 0.3, hidden_size: 64}, we will fix the optimizer to

'Adam' and the hidden_size to 64, and tune the training related hyperparameters in a narrowed search space as {learning_rate [0.005, 0.1, 0.05], weight_decay [0.0005, 0.005, 0.05], dropout [0.1, 0.3, 0.5]}. The number of configurations in the search spaces of different baseline methods are presented in Table 7, 8, 9, and 10. More details of hyperparameter configurations can be found in Appendix H.

Table 7: Number of hyperparameter configurations in the search spaces of baseline methods for basic node classification on homophilic and heterophilic graphs. The size of the C2F search space depends on the coarse search result.

| Basic Node Classiciation | Grid Search | Random Search | Coarse Search | C2F Search |
|---|---|---|---|---|
| MLP | 540 | 108 | 54 | 61 - 80 |
| GCN | 540 | 108 | 54 | 61 - 80 |
| SAGE | 540 | 108 | 54 | 61 - 80 |
| APPNP | 900 | 180 | 90 | 97 - 116 |
| GAT | 1620 | 324 | 162 | 169 - 188 |
| ChebNet | 1620 | 324 | 162 | 169 - 188 |
| H2GCN | 540 | 108 | 54 | 61 - 80 |
| SGC | 540 | 108 | 54 | 61 - 80 |
| GPRGNN | 1440 | 288 | 144 | 151 - 170 |
| MixHop | 540 | 108 | 54 | 61 - 80 |

Table 8: Number of hyperparameter configurations in the search spaces of baseline methods for node classification on large graphs. The size of the C2F search space depends on the coarse search result.

| Large Graphs | Grid Search | Random Search | Coarse Search | C2F Search |
|---|---|---|---|---|
| GCN | 54 | 12 | 6 | 13 - 23 |
| SAGE | 54 | 12 | 6 | 13 - 23 |
| APPNP | 72 | 16 | 8 | 15 - 25 |

Table 9: Number of hyperparameter configurations in the search spaces of baseline methods for node classification on long-range graphs. The size of the C2F search space depends on the coarse search result.

| Long-range Graphs | Grid Search | Random Search | Coarse Search | C2F Search |
|---|---|---|---|---|
| MLP | 48 | 10 | 5 | 12 - 16 |
| GCN | 48 | 10 | 5 | 12 - 16 |
| SAGE | 48 | 10 | 5 | 12 - 16 |
| APPNP | 72 | 16 | 8 | 15 - 19 |
| SGC | 48 | 10 | 5 | 12 - 16 |
| GPRGNN | 72 | 16 | 8 | 15 - 19 |
| MixHop | 48 | 10 | 5 | 12 - 16 |

Table 10: Number of hyperparameter configurations in the search spaces of baseline methods for link prediction tasks. The size of the C2F search space depends on the coarse search result.

| Link Prediction | Grid Search | Random Search | Coarse Search | C2F Search |
|---|---|---|---|---|
| MLP | 540 | 108 | 54 | 61 - 80 |
| GCN | 540 | 108 | 54 | 61 - 80 |
| SAGE | 540 | 108 | 54 | 61 - 80 |
| APPNP | 1620 | 324 | 162 | 169 - 188 |
| ChebNet | 1620 | 324 | 162 | 169 - 188 |

# E    DATASETS

## E.1    DATASET SOURCES

All datasets used in our experiments are publicly available and can be easily retrieved from the following sources.

**Cora, Citeseer, Pubmed**

```
https://pytorch-geometric.readthedocs.io/en/latest/generated/
torch_geometric.datasets.Planetoid.html#torch_geometric.datasets.
Planetoid
```

**Computers, Photo**

```
https://pytorch-geometric.readthedocs.io/en/latest/generated/
torch_geometric.datasets.Amazon.html#torch_geometric.datasets.
Amazon
```

**CS**

```
https://pytorch-geometric.readthedocs.io/en/latest/generated/
torch_geometric.datasets.Coauthor.html#torch_geometric.datasets.
Coauthor
```

**Actor**

```
https://pytorch-geometric.readthedocs.io/en/latest/generated/
torch_geometric.datasets.Actor.html#torch_geometric.datasets.Actor
```

**Wisconsin**

```
https://pytorch-geometric.readthedocs.io/en/latest/generated/
torch_geometric.datasets.WebKB.html#torch_geometric.datasets.WebKB
```

**Roman-Empire, Amazon-Ratings, Minesweeper, Tolokers**

```
https://pytorch-geometric.readthedocs.io/en/latest/generated/
torch_geometric.datasets.HeterophilousGraphDataset.html#torch_
geometric.datasets.HeterophilousGraphDataset
```

**Chameleon, Squirrel**

```
https://pytorch-geometric.readthedocs.io/en/latest/generated/
torch_geometric.datasets.WikipediaNetwork.html#torch_geometric.
datasets.WikipediaNetwork
```

**Flickr**

```
https://pytorch-geometric.readthedocs.io/en/latest/generated/
torch_geometric.datasets.Flickr.html#torch_geometric.datasets.
Flickr
```

**Reddit**

```
https://pytorch-geometric.readthedocs.io/en/latest/generated/
torch_geometric.datasets.Reddit.html#torch_geometric.datasets.
Reddit
```

**OGB-arXiv, OGB-Products**

```
https://ogb.stanford.edu/docs/nodeprop/
```

**PascalVOC-SP, COCO-SP**

```
https://github.com/vijaydwivedi75/lrgb
```

### E.2 DATASET STATISTICS

We provide statistics of 20 datasets used in our experiments in Table 11, 12, 13, 14, and 15.

Table 11: Datasets in basic node classification on homophilic graphs. We provide statistics of the number of nodes, edges, features, and node classes. "#" means "the number of".

| Dataset | #Nodes | #Egdes | #Features | #Node Classes |
|---|---|---|---|---|
| Cora | 2,708 | 5,429 | 1,433 | 7 |
| Citeseer | 3,327 | 4,372 | 3,703 | 6 |
| Pubmed | 19,717 | 44,338 | 500 | 3 |
| Computers | 13,752 | 491,722 | 767 | 10 |
| Photo | 7,487 | 126,530 | 745 | 8 |
| CS | 18,333 | 100,227 | 6805 | 15 |

Table 12: Datasets in basic node classification on heterophilic graphs. We provide statistics of the number of nodes, edges, features, and node classes. "#" means "the number of".

| Dataset | #Nodes | #Egdes | #Features | #Node Classes |
|---|---|---|---|---|
| Actor | 7,600 | 30,019 | 932 | 5 |
| Wisconsin | 251 | 515 | 1,703 | 5 |
| Roman-Empire | 22,662 | 32,927 | 300 | 18 |
| Amazon-Ratings | 24,492 | 93,050 | 300 | 5 |
| Minesweeper | 10,000 | 39,402 | 7 | 2 |
| Tolokers | 11,758 | 519,000 | 10 | 2 |

Table 13: Datasets in node classification on large graphs. We provide statistics of the number of nodes, edges, features, and node classes. "#" means "the number of".

| Dataset | #Nodes | #Egdes | #Features | #Node Classes |
|---|---|---|---|---|
| Flickr | 89,250 | 899,756 | 500 | 7 |
| Reddit | 232,965 | 114,615,892 | 602 | 41 |
| OGB-arXiv | 169,343 | 1,166,243 | 128 | 40 |
| OGB-Products | 2,449,029 | 61,859,140 | 100 | 47 |

Table 14: Datasets in node classification on long-range graphs. Long-range graph datasets contain multiple small graphs. We provide statistics of the number of graphs, all nodes, average nodes, edges, features, and node classes. "#" means "the number of".

| Dataset | #Graphs | #Nodes (all) | #Nodes (avg.) | #Egdes | #Features | #Node Classes |
|---|---|---|---|---|---|---|
| PascalVOC-SP | 11,355 | 5,443,545 | ~479.4 | 30,777,444 | 14 | 21 |
| COCO-SP | 123,286 | 58,793,216 | ~476.9 | 332,091,902 | 14 | 81 |

Table 15: Datasets in link prediction. We provide statistics of the number of nodes, edges, features, and link classes. "#" means "the number of".

| Dataset | #Nodes | #Egdes | #Features | #Link Classes |
|---|---|---|---|---|
| Cora | 2,708 | 10,556 | 1,433 | 2 |
| Citeseer | 3,327 | 9,104 | 3,703 | 2 |
| Chameleon | 2,277 | 36,101 | 2,325 | 2 |
| Squirrel | 5,201 | 270,962 | 2,089 | 2 |

### E.3 DATASET TRAIN/VAL/TEST SPLITS

For `Cora`, `Citeseer`, and `Pubmed` in node classification tasks, we adopt the public fixed train/test/val splits in (Yang et al., 2016). For `Computers`, `Photo`, and `CS`, we generate the splits with 20 nodes and 30 nodes per class in the training and validation sets and the rest in the test set. For heterophilic datasets, we adopt their original splits from (Pei et al., 2020) and (Platonov et al., 2023), and use the first split out of the 10 available splits for each dataset. This is because only one train/val/test split is considered in our experiments to easily compare the generated performance with the training performance over multiple runs. Large graph datasets are splited in the same way as in their original papers (Zeng et al., 2020; Hamilton et al., 2019; Hu et al., 2020a). The long-range datasets, `PascalVOC-SP` and `COCO-SP`, both contain multiple graphs. Hence, we adopt the train, validation, and test loaders from the implementation in Dwivedi et al. (2022). Besides, in consideration of time, for `COCO-SP`, we only use 10% of graphs in each loader for our experiment, and we ensure that the reduced loaders contain all node classes. Furthermore, the link prediction datasets are splited based on edges. Please refer to Appendix G.4 for details.

## F TARGET MODELS

In Table 16, we show details of the target models in our experiments. The original graph data is denoted as $\mathbf{X}^{(0)}$, and the output of layer $\ell$ is denoted as $\mathbf{X}^{(\ell)}$. In addition, $\mathbf{X}^{(\text{out})}$ is the output of the target model, and we only show how $\mathbf{X}^{(\text{out})}$ is computed when the last layer is different from the other layers. We use $\mathbf{W}^{(\ell)}$ to represent learnable linear operators in layer $\ell$, and $\sigma$ to represent the activation function. The adjacency matrix $\mathbf{A}$ can be normalized in implementation. We focus on the model-specific hyperparameters when designing the search spaces in our experiments. Please refer to the original papers for details of the other notations.

Table 16: Details of target models. We provide information of model layers, model-specific hyperparameters, and implementation sources.

| Model | Layer | Model-specific Hyperparameter |
|---|---|---|
| MLP | $\mathbf{X}^{(\ell+1)} = \sigma\big(\mathbf{X}^{(\ell)}\mathbf{W}^{(\ell)}\big)$ | - |
| GCN | $\mathbf{X}^{(\ell+1)} = \sigma\big(\mathbf{A}\mathbf{X}^{(\ell)}\mathbf{W}^{(\ell)}\big)$ | - |
| SAGE | $\mathbf{X}^{(\ell+1)} = \sigma\big(\mathbf{X}^{(\ell)}\mathbf{W}_1^{(\ell)} + \mathbf{A}\mathbf{X}^{(\ell)}\mathbf{W}_2^{(\ell)}\big)$ | - |
| APPNP | $\mathbf{X}^{(\ell+1)} = (1-\alpha)\mathbf{A}\mathbf{X}^{(\ell)} + \alpha\mathbf{X}^{(0)} \quad \mathbf{X}^{(\text{out})} = \sigma\big(\mathbf{X}^{(L)}\mathbf{W}\big)$ | Teleport probability $\alpha$ |
| GAT | $\mathbf{X}^{(\ell+1)} = \sigma\big(\mathbf{\Theta}^{(\ell)} \odot \mathbf{A}\mathbf{X}^{(\ell)}\mathbf{W}^{(\ell)}\big)$ | Number of heads |
| ChebNet | $\mathbf{X}^{(\ell+1)} = \sigma\big(\sum_{k=0}^{K} \mathcal{T}_k(\widetilde{\mathbf{L}})\mathbf{X}^{(\ell)}\mathbf{W}^{(\ell)}\big)$ | Chebyshev filter size K |
| H2GCN | $\mathbf{X}_{i,h\in H}^{(\ell+1)} = \text{AGGR}\{\mathbf{X}_j^{(\ell)} : j \in \mathcal{N}_h(i)\} \quad \mathbf{X}_i^{(\text{out})} = \text{Combine}(\mathbf{X}_i^{(0)}, \mathbf{X}_i^{(1)}, \cdots \mathbf{X}_i^{(L)})\mathbf{W}$ | Hop set $H$ |
| SGC | $\mathbf{X}^{(\text{out})} = \sigma\big(\mathbf{A}^h\mathbf{X}^{(0)}\mathbf{W}\big)$ | Number of hops $h$ |
| GPRGNN | $\mathbf{X}^{(\ell+1)} = \sum_{m=0}^{M} \gamma_m \mathbf{A}\mathbf{X}^{(\ell)}\mathbf{W}^{(\ell)}$ | GPR weights $\gamma_m$ |
| MixHop | $\mathbf{X}^{(\text{out})} = \text{Combine}_{h\in H}(\mathbf{A}^h\mathbf{X}^{(0)}\mathbf{W})$ | Hop set $H$ |

| Model | Implementation Source |
|---|---|
| MLP | https://pytorch.org/docs/stable/generated/torch.nn.Linear.html |
| GCN | https://pytorch-geometric.readthedocs.io/en/latest/modules/nn.html#convolutional-layers |
| SAGE | https://pytorch-geometric.readthedocs.io/en/latest/modules/nn.html#convolutional-layers |
| APPNP | https://pytorch-geometric.readthedocs.io/en/latest/modules/nn.html#convolutional-layers |
| GAT | https://pytorch-geometric.readthedocs.io/en/latest/modules/nn.html#convolutional-layers |
| ChebNet | https://pytorch-geometric.readthedocs.io/en/latest/modules/nn.html#convolutional-layers |
| H2GCN | https://github.com/GemsLab/H2GCN |
| SGC | https://pytorch-geometric.readthedocs.io/en/latest/modules/nn.html#convolutional-layers |
| GPRGNN | https://github.com/jianhao2016/GPRGNN |
| MixHop | https://pytorch-geometric.readthedocs.io/en/latest/modules/nn.html#convolutional-layers |

## G  TASKS AND RELEVANT DETAILS

### G.1  BASIC NODE CLASSIFICATION

**Task description.** The basic node classification is a semi-supervised task on small-scale homophilic and heterophilic graphs. During training, the GNN has access to the features of all nodes and the graph adjacency matrix, but the loss function is computed only based on the training node labels. Likewise, validation and test are conducted with the validation and test node labels. Training, validation, and test labels are obtained with masks.

**Target model architectures.** We use a 2-convolutional layer structure for most target models following the common practice. Similarly, for MLP, we use 2 linear layers, which often leads to good performance on heterophilic graphs (Xu et al., 2019). For SGC and MixHop, which are designed to have only one graph convolutional layer, we adjust by setting 2 hops for SGC and {0, 1, 2} hops for MixHop. H2GCN also uses a hop set of {0, 1, 2}, but like most target models, we set the number of layers to 2. The ReLU activation function is used in all layers except for the last layer when applicable. Importantly, all target models have simple architectures without using advanced techniques like residual connections or layer normalization.

### G.2  NODE CLASSIFICATION ON LARGE GRAPHS

**Task description.** The node classification on large graphs is the same as the basic node classification except that the graphs are very large, which may cause memory issues and slow computation. To solve this problem, we apply graph clustering as motivated by (Chiang et al., 2019). Graphs are clustered into multiple densely connected subgraphs by minimizing the number of edges cut between clusters. The number of clusters is set to 32 for `Flickr` and `OGB-arXiv` and 960 for `Reddit` and `OGB-Products`. Each cluster has its own train/validation/test masks. During training, we loop through all clusters iteratively via a cluster loader in each epoch. Validation and testing are implemented with the cluster loader without shuffling.

**Target model architectures.** We use GCN, SAGE, and APPNP as target models. GCN and SAGE adopt the 2-convolutional layer structure with the hidden size tuned in [64, 128, 256]. APPNP has 2 graph convolutional layers followed by a linear layer to map the graph representation formed by graph convolutional layers to node classes. The ReLU activation function is used in all layers except for the last layer when applicable.

### G.3  NODE CLASSIFICATION ON LONG-RANGE GRAPHS

**Task description.** The node classification on long-range graphs is different from the previous two tasks as there are multiple small graphs in long-range graph datasets. These small graphs are divided into train/validation/test sets. Therefore, the node classification on long-range graphs is a supervised task instead of a semi-supervised task. In addition, the node classes are highly imbalanced in the datasets used in our experiments. For example, in `PascalVOC-SP`, there are 21 classes with more than 70% of nodes in class 0. Accordingly, we use the weighted cross entropy loss for training as in (Dwivedi et al., 2022), which applies higher weights to uncommon classes. In data preprocessing, we construct train/validation/test loaders with batch sizes 128, 500, and 500, respectively. The small graphs in each batch are connected into a large batch graph such that they can be processed by target models in parallel. We train the target model with 200 epochs with 50 batches in each epoch and then evaluate the model with all batches in the validation and test loaders.

**Target model architectures.** Since long-range graphs usually rely on very deep GNN architectures, we set the number of layers as a hyperparameter in the search spaces. For most target models, we adjust the number of layers in [8, 10] and the hidden size in [192, 256]. For SGC and MixHop, since they are designed to have only one graph convolutional layer, we tune the number of hops of SGC in [8, 10], and the hop set of MixHop in [{6, 8, 10}, {8, 10}]. Furthermore, considering the taget models are very deep, we adopt the GELU activation function and residual connections following the experiment settings in (Tönshoff et al., 2024) as well as layer normalization.

## G.4 LINK PREDICTION

**Task description.** The link prediction is a binary classification task that predicts the existence of edges between pairs of nodes (Kipf & Welling, 2016). If there exists an edge between a pair of nodes, then the edge is called a "positive link", which is corresponding to class 1. In contrast, if there is no edge between a pair of nodes, then it is treated as a "negative link", which is corresponding to class 0. The prediction requires two steps: (1) an encoder to construct node embeddings based on the positive links; (2) a decoder to aggregate embeddings of target node pairs by taking inner products. An illustration of the prediction process is provided in Figure 11.

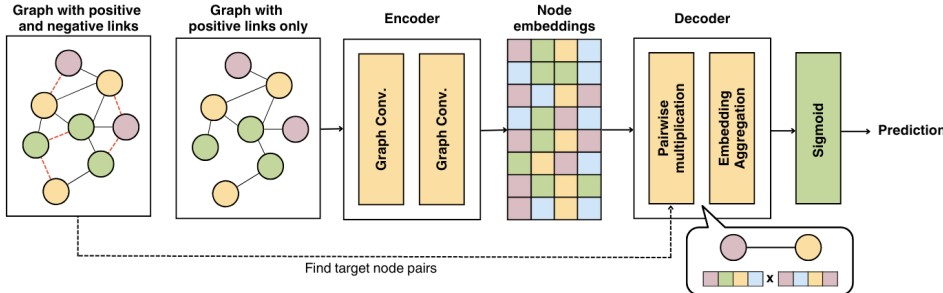

Figure 11: Illustration of the link prediction process. This figure is adapted from a nice tutorial in `https://github.com/tomonori-masui/graph-neural-networks/blob/main/gnn_pyg_implementations.ipynb`.

The train/validation/test split is conducted on the edge level. More specifically, we randomly select 5% edges as validation edges and 10% edges as test edges, and the rest will be training edges. The selected edges are the positive links, and the negative links will be randomly sampled from disconneted node pairs. The number of negative egdes is set to be the same as the number of positive edges, which enables us to use accuracy as the metric. We present how the training, validation, and testing edges are used in Figure 12.

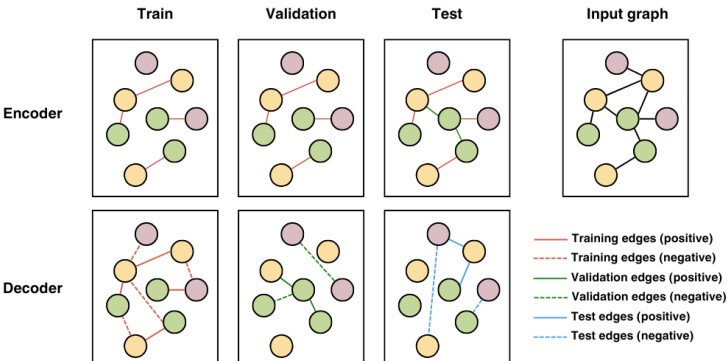

Figure 12: Edges used in the encoder and decoder during training, validation, and testing. This figure is adapted from a nice tutorial in `https://github.com/tomonori-masui/graph-neural-networks/blob/main/gnn_pyg_implementations.ipynb`.

**Target model architectures.** All target models are used as the encoder in the link prediction task. Target GNNs all adopt 2-convolutional layer structure, while the MLP is composed of two linear layers. For APPNP, we apply a linear layer after the 2 graph convolutional layers to reduce the embedding dimension. The ReLU activation function is applied in the encoder when applicable. The output dimension of the encoder is the same as the hidden size of the target model, and is treated as a hyperparameter that requires tuning. The decoders are the same for all target models, taking inner products of target node pairs and then passing to the Sigmoid activation function for final prediction.

# H HYPERPARAMETER TUNING SEARCH SPACE

## H.1 FULL SEARCH SPACES - BASIC NODE CLASSIFICATION

**Training-related Hyperparameters**

optimizer: ['SGD', 'Adam']

learning_rate: [0.005, 0.01, 0.05, 0.1, 0.5, 1]

weight_decay: [0.0005, 0.005, 0.05]

dropout: [0.1, 0.3, 0.5, 0.7, 0.9]

scheduler: ['MultiStepLR']

scheduler_milestone: [[100, 125, 150, 175]]

scheduler_gamma: [0.2]

**Model-related Hyperparameters**

MLP: hidden_size [16, 32, 64]

GCN: hidden_size [16, 32, 64]

SAGE: hidden_size [16, 32, 64]

APPNP: teleport_probability [0.1, 0.3, 0.5, 0.7, 0.9]

GAT: hidden_size [16, 32, 64]; num_heads [4, 8, 12]

ChebNet: hidden_size [16, 32, 64]; filter_size [1, 2, 3]

H2GCN: hidden_size [16, 32, 64]

SGC: num_hops [1, 2, 3]

GPRGNN: init ['Random', 'PPR'], GPR_alpha [0.1, 0.2, 0.5, 0.9]

MixHop: hidden_size [16, 32, 64]; power_set [{0, 1, 2}]

## H.2 FULL SEARCH SPACES - NODE CLASSIFICATION ON LARGE GRAPHS

**Training-related Hyperparameters**

optimizer: ['Adam']

learning_rate: [0.0005, 0.005, 0.05]

weight_decay: [0.0005, 0.005]

dropout: [0.1, 0.3, 0.7]

scheduler: ['MultiStepLR']

scheduler_milestone: [[150, 170]]

scheduler_gamma: [0.2]

**Model-related Hyperparameters**

GCN: hidden_size [64, 128, 256]

SAGE: hidden_size [64, 128, 256]

APPNP: teleport_probability [0.1, 0.3, 0.5, 0.9]

### H.3 Full Search Spaces - Node Classification on Long-range Graphs

**Training-related Hyperparameters**

optimizer: ['AdamW']

learning_rate: [0.0005, 0.001, 0.005]

weight_decay: [0.0005, 0.005]

dropout: [0.1, 0.3]

scheduler: None

**Model-related Hyperparameters**

MLP: hidden_size [192, 256], num_layers [8, 10]

GCN: hidden_size [192, 256], num_layers [8, 10]

SAGE: hidden_size [192, 256], num_layers [8, 10]

APPNP: hidden_size [256], num_layers [8, 10], teleport_probability [0.1, 0.5, 0.9]

SGC: hidden_size [192, 256], num_hops [8, 10]

GPRGNN: hidden_size [256], num_layers [8, 10], init ['PPR'], GPR_alpha [0.1, 0.5, 0.9]

MixHop: hidden_size [192, 256], power_set [{6, 8, 10}, {8, 10}]

### H.4 Full Search Spaces - Link Prediction

**Training-related Hyperparameters**

optimizer: ['SGD', 'Adam']

learning_rate: [0.005, 0.01, 0.05, 0.1, 0.5, 1]

weight_decay: [0.0005, 0.005, 0.05]

dropout: [0.1, 0.3, 0.5, 0.7, 0.9]

scheduler: ['MultiStepLR']

scheduler_milestone: [[100, 125, 150, 175]]

scheduler_gamma: [0.2]

**Model-related Hyperparameters**

MLP: hidden_size [16, 32, 64]

GCN: hidden_size [16, 32, 64]

SAGE: hidden_size [16, 32, 64]

APPNP: hidden_size [16, 32, 64], teleport_probability [0.1, 0.5, 0.9]

ChebNet: hidden_size [16, 32, 64]; filter_size [1, 2, 3]

# I SUPPLEMENTARY EXPERIMENT RESULTS AND DISCUSSIONS

## I.1 MORE RESULTS OF NODE CLASSIFICATION ON HOMOPHILIC GRAPHS

Table 17: More results of basic node classification on heterophilic graphs. We report the average test accuracy (%) and the standard deviation of models selected by validation over 10 training or generation runs. The best results are marked in **bold**.

| Datasets | Pubmed (Homophily) | | | | | Computers (Homophily) | | | | |
|---|---|---|---|---|---|---|---|---|---|---|
| Models | Grid | Random | Coarse | C2F | GNN-Diff | Grid | Random | Coarse | C2F | GNN-Diff |
| MLP | 73.83 ± 0.31 | 73.79 ± 0.45 | 74.07 ± 0.87 | 74.11 ± 0.58 | **74.73 ± 0.10** | 66.40 ± 2.18 | 66.79 ± 1.62 | 67.34 ± 0.76 | 66.39 ± 2.01 | **67.57 ± 0.16** |
| GCN | 79.27 ± 0.40 | 79.09 ± 0.47 | 78.70 ± 1.09 | 79.09 ± 0.47 | **79.30 ± 0.31** | **82.42 ± 0.78** | 80.69 ± 1.42 | 80.35 ± 1.39 | 79.07 ± 0.42 | 82.39 ± 0.29 |
| SAGE | 77.26 ± 0.34 | 77.55 ± 0.35 | 77.61 ± 0.41 | 77.27 ± 0.62 | **77.65 ± 0.11** | **80.19 ± 1.89** | 78.92 ± 1.87 | 78.69 ± 3.06 | **80.19 ± 1.89** | 79.80 ± 0.22 |
| APPNP | 79.15 ± 0.70 | 79.15 ± 0.61 | 78.72 ± 0.54 | 77.55 ± 0.89 | **79.28 ± 0.20** | 82.77 ± 1.55 | 82.35 ± 1.58 | 83.30 ± 0.81 | 83.37 ± 1.62 | **83.93 ± 0.58** |
| GAT | 78.45 ± 0.69 | 78.33 ± 0.64 | 78.08 ± 0.43 | **78.46 ± 0.68** | 78.40 ± 0.03 | 82.70 ± 0.85 | 82.43 ± 0.93 | 82.06 ± 0.58 | 82.94 ± 0.96 | **83.16 ± 0.44** |
| ChebNet | 78.29 ± 0.71 | 77.30 ± 0.59 | 77.82 ± 0.35 | 77.45 ± 0.70 | **78.31 ± 0.58** | **73.76 ± 2.77** | 69.58 ± 1.82 | 69.56 ± 1.81 | 69.57 ± 1.78 | 70.12 ± 0.66 |
| H2GCN | 79.18 ± 0.38 | 79.46 ± 0.12 | 78.99 ± 0.42 | 79.01 ± 0.21 | **79.76 ± 0.47** | 77.29 ± 1.36 | 75.21 ± 3.50 | 73.79 ± 4.34 | 75.02 ± 2.39 | **77.81 ± 0.92** |
| SGC | 78.50 ± 0.16 | 77.44 ± 0.77 | 78.32 ± 0.04 | 78.91 ± 0.57 | **79.10 ± 0.19** | 80.37 ± 1.65 | 80.91 ± 1.56 | 81.88 ± 0.91 | 81.79 ± 0.63 | **82.26 ± 0.58** |
| GPRGNN | 79.56 ± 0.56 | 79.35 ± 0.42 | 79.19 ± 0.83 | 79.22 ± 0.83 | **79.69 ± 0.08** | 82.42 ± 0.60 | 82.79 ± 1.40 | 82.58 ± 0.58 | 82.58 ± 0.58 | **82.80 ± 0.30** |
| MixHop | 76.96 ± 0.94 | 76.42 ± 0.34 | 76.91 ± 0.80 | 76.96 ± 0.69 | **78.03 ± 0.21** | 73.62 ± 1.44 | 73.00 ± 0.96 | 72.76 ± 1.57 | 72.57 ± 0.96 | **75.64 ± 0.13** |

| Datasets | Photo (Homophily) | | | | | CS (Homophily) | | | | |
|---|---|---|---|---|---|---|---|---|---|---|
| Models | Grid | Random | Coarse | C2F | GNN-Diff | Grid | Random | Coarse | C2F | GNN-Diff |
| MLP | 80.61 ± 0.64 | 80.46 ± 0.69 | 80.45 ± 0.69 | 80.45 ± 0.69 | **81.69 ± 0.02** | **88.32 ± 0.60** | 87.45 ± 0.91 | 86.90 ± 1.28 | 87.59 ± 0.95 | 87.63 ± 0.01 |
| GCN | 90.83 ± 0.45 | 90.81 ± 0.37 | 90.96 ± 0.47 | 91.10 ± 1.57 | **91.41 ± 0.01** | 91.34 ± 0.19 | 91.04 ± 0.34 | 91.25 ± 0.14 | 91.17 ± 0.14 | **91.35 ± 0.02** |
| SAGE | 90.32 ± 0.55 | 90.27 ± 0.70 | 89.62 ± 0.91 | 88.35 ± 1.35 | **90.36 ± 0.19** | 90.98 ± 0.33 | 90.95 ± 0.32 | 90.75 ± 0.58 | 90.91 ± 0.17 | **91.01 ± 0.40** |
| APPNP | 90.43 ± 0.77 | 90.34 ± 0.94 | 90.12 ± 0.45 | 90.29 ± 0.89 | **90.99 ± 0.50** | 92.13 ± 0.25 | 92.09 ± 0.30 | 91.73 ± 0.20 | 92.13 ± 0.25 | **92.30 ± 0.02** |
| GAT | 90.25 ± 0.92 | 90.24 ± 0.71 | 89.71 ± 0.89 | 90.11 ± 1.06 | **91.01 ± 0.48** | 90.35 ± 0.46 | 89.81 ± 0.78 | 89.52 ± 0.9 | 89.73 ± 1.14 | **90.57 ± 0.08** |
| ChebNet | **87.73 ± 2.87** | 84.88 ± 2.13 | 83.85 ± 3.07 | 86.28 ± 1.81 | 85.98 ± 0.13 | 91.79 ± 0.53 | 91.76 ± 0.65 | 91.55 ± 0.39 | 91.82 ± 0.35 | **91.90 ± 0.04** |
| H2GCN | 90.30 ± 0.80 | 89.99 ± 0.60 | 89.17 ± 1.47 | 90.29 ± 0.76 | **90.83 ± 0.44** | **92.24 ± 0.32** | 92.11 ± 0.13 | 91.27 ± 0.41 | 92.07 ± 0.24 | **92.24 ± 0.04** |
| SGC | 90.27 ± 0.60 | 90.47 ± 0.52 | 90.46 ± 0.56 | 90.24 ± 1.20 | **90.65 ± 0.17** | **91.59 ± 0.12** | 90.99 ± 0.16 | 91.37 ± 0.23 | 91.02 ± 0.38 | 91.46 ± 0.09 |
| GPRGNN | 91.18 ± 0.38 | 90.28 ± 1.26 | 90.07 ± 1.20 | 91.24 ± 0.59 | **91.26 ± 0.28** | **90.94 ± 0.44** | 90.81 ± 0.44 | 90.66 ± 0.53 | 90.75 ± 2.01 | 90.93 ± 0.14 |
| MixHop | **86.31 ± 1.31** | 85.04 ± 1.17 | 84.47 ± 0.73 | 84.67 ± 1.39 | 86.12 ± 0.60 | 92.10 ± 0.33 | 92.09 ± 0.33 | 91.05 ± 0.72 | 91.82 ± 0.51 | **92.23 ± 0.05** |

## I.2 MORE RESULTS OF NODE CLASSIFICATION ON HETEROPHILIC GRAPHS

Table 18: More results of basic node classification on homophilic graphs. We report the average test accuracy (%) and the standard deviation of models selected by validation over 10 training or generation runs. The best results are marked in **bold**.

| Datasets | Roman-Empire (Heterophily) | | | | | Amazon-Ratings (Heterophily) | | | | |
|---|---|---|---|---|---|---|---|---|---|---|
| Models | Grid | Random | Coarse | C2F | GNN-Diff | Grid | Random | Coarse | C2F | GNN-Diff |
| MLP | **66.19 ± 0.17** | 65.32 ± 0.36 | 65.27 ± 0.29 | 65.27 ± 0.29 | 66.11 ± 0.04 | **40.20 ± 0.52** | 38.52 ± 0.68 | 38.56 ± 0.16 | 39.60 ± 0.35 | 39.82 ± 0.20 |
| GCN | **48.87 ± 0.20** | 47.44 ± 0.57 | 47.43 ± 0.42 | 47.70 ± 0.51 | 48.43 ± 0.17 | 43.10 ± 0.13 | 41.52 ± 0.28 | 41.16 ± 0.16 | 43.11 ± 0.33 | **43.19 ± 0.14** |
| SAGE | **78.12 ± 0.22** | 77.32 ± 0.29 | 76.91 ± 0.32 | 76.91 ± 0.32 | 77.65 ± 0.12 | 43.23 ± 0.52 | 43.27 ± 0.51 | 42.78 ± 0.57 | 43.03 ± 0.52 | **43.32 ± 0.06** |
| APPNP | 58.23 ± 0.19 | 58.14 ± 0.21 | 57.71 ± 0.06 | 58.23 ± 0.19 | **58.36 ± 0.34** | 38.79 ± 0.17 | 39.03 ± 0.13 | 38.86 ± 0.63 | 38.86 ± 0.63 | **39.26 ± 0.16** |
| GAT | **56.86 ± 0.58** | 55.53 ± 1.26 | 55.31 ± 0.73 | 56.10 ± 1.00 | 55.40 ± 0.04 | 44.74 ± 0.30 | 44.41 ± 0.46 | 44.39 ± 0.37 | 44.42 ± 0.35 | **44.77 ± 0.04** |
| ChebNet | 80.05 ± 0.27 | 79.09 ± 0.40 | 78.63 ± 0.21 | 80.05 ± 0.27 | **80.06 ± 0.04** | **44.56 ± 0.16** | 42.87 ± 0.35 | 43.96 ± 0.28 | 43.93 ± 0.29 | 44.04 ± 0.18 |
| H2GCN | 63.63 ± 2.30 | 62.96 ± 0.65 | 62.12 ± 1.19 | 63.32 ± 1.27 | **63.94 ± 0.07** | **43.24 ± 0.27** | 43.09 ± 0.26 | 42.65 ± 0.08 | 42.76 ± 0.29 | 42.83 ± 0.15 |
| SGC | **39.10 ± 0.17** | 39.02 ± 0.12 | 38.28 ± 0.05 | 39.01 ± 0.12 | 38.47 ± 0.16 | **39.96 ± 0.53** | 39.70 ± 0.27 | 39.56 ± 0.18 | 39.37 ± 0.17 | 39.60 ± 0.12 |
| GPRGNN | 71.70 ± 0.31 | 71.63 ± 0.46 | 69.82 ± 0.42 | 69.81 ± 0.44 | **71.76 ± 0.07** | 44.13 ± 0.38 | 44.06 ± 0.48 | 43.95 ± 0.39 | 44.17 ± 0.52 | **44.24 ± 0.17** |
| MixHop | 77.99 ± 0.14 | 76.95 ± 0.31 | 74.10 ± 0.13 | **79.01 ± 0.16** | 74.94 ± 0.13 | **43.03 ± 0.32** | 42.61 ± 0.37 | 41.25 ± 0.16 | 41.64 ± 0.25 | 42.29 ± 0.07 |

| Datasets | Minesweeper (Heterophily) | | | | | Tolokers (Heterophily) | | | | |
|---|---|---|---|---|---|---|---|---|---|---|
| Models | Grid | Random | Coarse | C2F | GNN-Diff | Grid | Random | Coarse | C2F | GNN-Diff |
| MLP | **80.00 ± 0.00** | 79.97 ± 0.08 | 79.94 ± 0.12 | 79.72 ± 0.09 | **80.00 ± 0.00** | 78.22 ± 0.04 | 78.18 ± 0.03 | 78.16 ± 0.01 | 78.17 ± 0.02 | **78.24 ± 0.05** |
| GCN | 80.26 ± 0.10 | 80.26 ± 0.12 | 80.19 ± 0.10 | 80.28 ± 0.14 | **80.28 ± 0.07** | 78.65 ± 0.07 | 78.67 ± 0.09 | 78.63 ± 0.10 | 78.67 ± 0.05 | **78.73 ± 0.08** |
| SAGE | 85.63 ± 0.20 | 85.56 ± 0.21 | 85.23 ± 0.27 | 85.63 ± 0.20 | **85.78 ± 0.32** | 78.25 ± 0.20 | 78.44 ± 0.12 | 78.34 ± 0.21 | 78.47 ± 0.11 | **78.51 ± 0.06** |
| APPNP | 80.20 ± 0.07 | 80.11 ± 0.13 | 79.96 ± 0.08 | 80.01 ± 0.13 | **80.26 ± 0.02** | 78.62 ± 0.80 | 78.61 ± 0.04 | 78.56 ± 0.12 | **78.63 ± 0.04** | 78.63 ± 0.03 |
| GAT | 81.98 ± 0.62 | 81.34 ± 0.53 | 81.18 ± 0.37 | 81.81 ± 0.73 | **82.31 ± 0.06** | 79.73 ± 0.50 | 79.50 ± 0.61 | 78.34 ± 0.59 | 78.35 ± 0.87 | **79.80 ± 0.12** |
| ChebNet | 86.82 ± 0.29 | 86.64 ± 0.38 | 84.66 ± 1.26 | 86.86 ± 0.20 | **86.89 ± 0.27** | 79.43 ± 0.60 | 78.83 ± 0.49 | 78.56 ± 0.13 | 78.68 ± 0.14 | **79.66 ± 0.19** |
| H2GCN | 83.59 ± 1.03 | 83.31 ± 0.33 | 82.67 ± 0.92 | 83.61 ± 0.49 | **83.68 ± 0.32** | **78.83 ± 0.52** | 78.56 ± 0.13 | 78.37 ± 0.25 | 78.51 ± 0.09 | 78.80 ± 0.07 |
| SGC | 81.51 ± 0.11 | 81.61 ± 0.17 | 81.46 ± 0.20 | 81.55 ± 0.22 | **81.63 ± 0.24** | 78.51 ± 0.02 | 78.49 ± 0.03 | 78.47 ± 0.09 | **78.63 ± 0.04** | 78.57 ± 0.03 |
| GPRGNN | 83.92 ± 0.52 | 83.92 ± 0.52 | 83.94 ± 0.56 | 83.94 ± 0.56 | **83.97 ± 0.03** | **78.38 ± 0.27** | 78.36 ± 0.21 | 78.28 ± 0.16 | 78.37 ± 0.22 | 78.28 ± 0.06 |
| MixHop | 83.52 ± 0.14 | 82.75 ± 0.20 | 83.46 ± 0.21 | 83.86 ± 0.14 | **83.92 ± 0.12** | 79.51 ± 0.15 | 78.7 ± 0.33 | 79.22 ± 0.24 | 79.51 ± 0.15 | **79.54 ± 0.41** |

### I.3 EXTENSION TO GRAPH-LEVEL TASKS AND EXPERIMENT RESULTS

**GNN-Diff for Graph Classification and Regression**

Graph classification is a task where we try to predict the label of an entire graph from a collection of graphs. This is different from node classification, where we predict labels for individual nodes within a graph with representations learned for each node. In graph classification, we need to create a global representation for the whole graph, which captures information from all the nodes in the graph, in order to make the final prediction. To achieve this, GNNs for graph classification usually construct node-level representations first, then apply a graph pooling layer to integrate all node information (Liu et al., 2022). The graph pooling process can be as simple as taking the sum or average of all node representations, or employing more advanced methods such as differentiable pooling (Ying et al., 2018) and top-k pooling (Gao & Ji, 2019). Graph regression is very similar to graph classification, with the main difference being the type of loss function and evaluation metrics.

Figure 13: GFE architecture for graph classification and graph regression. The graph condition is designed to incorporate both data and graph structural information for GNN parameter generation.

The extension of GNN-Diff to graph classification and regression involves the change of GFE (see Figure 13). The input of GFE is a set of graphs. We denote the feature matrix and adjacency matrix of each graph as $\mathbf{X}_i$ and $\mathbf{A}_i$ for $i = 1, 2, ..., I$. In the GFE encoder, each graph is processed by a graph convolutional layer, followed by a mean pooling layer as

$$\eta_i = \text{G-Encoder}(\mathbf{X}_i, \mathbf{A}_i) = \text{POOL}(\text{GNN}(\mathbf{X}_i, \mathbf{A}_i)) \in \mathbb{R}^{D_p}. \tag{9}$$

Note that the node dimension is aggregated into 1 by the pooling layer. In the following experiments, we apply a single GCN layer for the graph convolution. The GFE decoder is a single linear layer. Moreoever, the loss function needs to adjusted based on the task. Normally, we may choose the cross entropy loss for graph classification, and the mean squared error loss for graph regression. Finally, to construct the graph condition, we take the mean of the global representations of all input graphs as $c_{\mathcal{G}} = \frac{1}{I} \sum_{i=1}^{I} \eta_i \in \mathbb{R}^{D_p}$.

**Experiments Results of Graph Classification**

Table 19: Fake news datasets for graph classification experiments. Each dataset contain multiple graphs. We provide statistics of the number of graphs, all nodes, average nodes, edges, features, and node classes. "#" means "the number of".

| Dataset | #Graphs | #Nodes (all) | #Nodes (avg.) | #Edges | #Features | #Classes |
|---------|---------|--------------|---------------|--------|-----------|----------|
| Politifact | 314 | 41,054 | 131 | 40,740 | 300 | 2 |
| Gossipcop | 5464 | 314,262 | 58 | 308,798 | 300 | 2 |

To evaluate the effectiveness of GNN-Diff on graph-level tasks, we conduct graph classification experiments for fake news detection using two datasets: Politifact and Gossipcop (Shu et al., 2020; Dou et al., 2021). Politifact contains political news, while Gossipcop focuses on entertainment news. Both datasets consist of tree-structured graphs derived from Twitter, where the task is to classify each graph as either fake or real news. Node features are derived from embeddings of Twitter users' historical tweets. More details of the datasets can be found in Figure 19.

Table 20: Experiment results of graph classification. We report the average test accuracy (%) and the standard deviation of models selected by validation over 10 training or generation runs. The best results are marked in **bold**.

| **Politifact** | | | | | |
|---|---|---|---|---|---|
| Models | Grid | Random | Coarse | C2F | GNN-Diff |
| GCN | $80.14 \pm 1.94$ | $79.23 \pm 1.95$ | $78.42 \pm 1.87$ | $77.19 \pm 1.22$ | $\mathbf{80.45 \pm 1.12}$ |
| SAGE | $76.24 \pm 3.47$ | $76.60 \pm 2.71$ | $77.10 \pm 0.91$ | $77.60 \pm 2.85$ | $\mathbf{78.28 \pm 1.71}$ |
| **Gossipcop** | | | | | |
| Models | Grid | Random | Coarse | C2F | GNN-Diff |
| GCN | $97.14 \pm 0.08$ | $96.85 \pm 0.16$ | $96.58 \pm 0.23$ | $97.06 \pm 0.09$ | $\mathbf{97.62 \pm 0.09}$ |
| SAGE | $\mathbf{96.73 \pm 0.24}$ | $96.72 \pm 0.06$ | $96.59 \pm 0.14$ | $96.84 \pm 0.14$ | $\mathbf{96.73 \pm 0.07}$ |

We use two target models, GCN and SAGE, and follow the data split and model implementation from the `PyTorch_Geometric` example[1]. The hyperparameter search space is as follows:

optimizer: ['SGD', 'Adam']

learning_rate: [0.005, 0.01, 0.05, 0.1, 0.5]

weight_decay: [0.0005, 0.005]

dropout: [0.0, 0.1, 0.3]

GCN_hidden_size [16, 32, 64]

SAGE_hidden_size [16, 32, 64].

The experimental results in Table 20 demonstrate that GNN-Diff consistently surpasses the baseline methods across both datasets—`Politifact` and `Gossipcop`—in the graph classification task for fake news detection. Specifically, for the `Politifact` dataset, GNN-Diff achieves the highest accuracy for both GCN and SAGE. Similarly, on the `Gossipcop` dataset, GNN-Diff maintains its superiority with GCN and SAGE, matching or exceeding the performance of other baselines while achieving the lowest variance. These results underscore GNN-Diff's robustness and effectiveness in addressing the graph classification challenges in the context of fake news detection, particularly in achieving high accuracy with consistent reliability.

---

[1]`https://github.com/pyg-team/pytorch_geometric/blob/master/examples/upfd.py`

## I.4 BAYESIAN OPTIMIZATION VS. GNN-DIFF

In our main experiments, we compare GNN-Diff with some traditional search methods, including coarse search, random search and grid search, and a more advanced method, coarse-to-fine (C2F) search, to efficiently reduce the search space based on the coarse search results. Here we consider another advanced hyperparameter tuning method, Bayesian optimization, which is less commonly used in GNN-related works but a very useful tool in the general machine learning field (Snoek et al., 2012; 2015). We implement the Bayesian search with Optuna (Akiba et al., 2019). Optuna is a hyperparameter tuning framework that by default adopts Tree-structured Parzen Estimators (TPEs), a probabilistic model-based optimization method that falls under the umbrella of Bayesian optimization. We compare the Bayesian search and GNN-Diff with basic node classification tasks on 6 homophilic and 6 heterophilic graphs (see Table 21). The Bayesian search space is almost the same as in Section H.1 except that we employ continuous range for the learning rate, weight decay, dropout and also the alphas in APPNP and GPRGNN to better suit the Bayesian hyperparameter tuning framework.

Table 21: Comparison between Bayesian search and GNN-Diff. The average test accuracy (%) and the corresponding standard deviation are reported. The best results are marked in **bold**.

| **Basic Node Classification - Homophilic Graphs** | | | | | |
|---|---|---|---|---|---|
| Datasets | Cora | | Citeseer | | Pubmed | |
| Models | Bayesian | GNN-Diff | Bayesian | GNN-Diff | Bayesian | GNN-Diff |
| MLP | 58.73 ± 0.78 | **59.47 ± 0.43** | 57.32 ± 1.48 | **58.72 ± 0.84** | 72.19 ± 1.98 | **74.73 ± 0.10** |
| GCN | 81.09 ± 0.37 | **82.33 ± 0.17** | 70.34 ± 0.48 | **72.37 ± 0.29** | 79.24 ± 0.22 | **79.30 ± 0.31** |
| SAGE | 80.43 ± 0.34 | **80.60 ± 0.15** | 70.40 ± 0.46 | **70.45 ± 0.14** | 76.87 ± 0.36 | **77.65 ± 0.11** |
| APPNP | 78.99 ± 1.06 | **82.51 ± 0.29** | 68.60 ± 1.15 | **71.44 ± 0.17** | 76.29 ± 0.90 | **79.28 ± 0.20** |
| GAT | 81.46 ± 1.25 | **81.69 ± 0.10** | 70.24 ± 0.39 | **71.50 ± 0.09** | 76.95 ± 0.61 | **78.40 ± 0.03** |
| ChebNet | 80.96 ± 0.76 | **82.05 ± 0.55** | 69.97 ± 0.58 | **71.65 ± 0.27** | 77.35 ± 0.80 | **78.31 ± 0.58** |
| H2GCN | 81.37 ± 1.89 | **82.17 ± 0.12** | 68.12 ± 0.91 | **71.78 ± 0.25** | 79.07 ± 0.81 | **79.76 ± 0.47** |
| SGC | 81.94 ± 0.24 | **82.10 ± 0.24** | 71.52 ± 2.77 | **72.10 ± 0.18** | 78.80 ± 0.27 | **79.10 ± 0.19** |
| GPRGNN | 81.51 ± 1.32 | **81.79 ± 0.20** | 71.12 ± 0.26 | **71.86 ± 0.19** | 79.52 ± 0.40 | **79.69 ± 0.08** |
| MixHop | 78.42 ± 0.93 | **80.32 ± 0.71** | 69.74 ± 0.72 | **71.50 ± 0.49** | 76.71 ± 0.25 | **78.03 ± 0.21** |
| Datasets | Computers | | Photo | | CS | |
| Models | Bayesian | GNN-Diff | Bayesian | GNN-Diff | Bayesian | GNN-Diff |
| MLP | 65.82 ± 1.42 | **67.57 ± 0.16** | 80.70 ± 0.91 | **81.69 ± 0.02** | 87.62 ± 0.28 | **87.63 ± 0.01** |
| GCN | 82.24 ± 0.91 | **82.39 ± 0.29** | 90.47 ± 0.69 | **91.41 ± 0.01** | 90.80 ± 0.12 | **91.35 ± 0.02** |
| SAGE | 79.45 ± 1.30 | **79.80 ± 0.22** | 90.33 ± 0.80 | **90.36 ± 0.19** | 90.95 ± 0.31 | **91.01 ± 0.40** |
| APPNP | 83.01 ± 1.55 | **83.93 ± 0.58** | 90.66 ± 1.17 | **90.99 ± 0.50** | 91.61 ± 0.52 | **92.30 ± 0.02** |
| GAT | 82.00 ± 1.02 | **83.16 ± 0.44** | 90.54 ± 1.46 | **91.01 ± 0.48** | 90.13 ± 0.33 | **90.57 ± 0.08** |
| ChebNet | **77.37 ± 1.74** | 70.12 ± 0.66 | **86.27 ± 1.90** | 85.98 ± 0.13 | 91.78 ± 0.42 | **91.90 ± 0.04** |
| H2GCN | 77.58 ± 1.27 | **77.81 ± 0.92** | 87.54 ± 2.60 | **90.83 ± 0.44** | 91.12 ± 0.74 | **92.24 ± 0.04** |
| SGC | 82.24 ± 0.12 | **82.26 ± 0.58** | **90.71 ± 0.13** | 90.65 ± 0.17 | 91.00 ± 0.43 | **91.46 ± 0.09** |
| GPRGNN | 78.97 ± 2.25 | **82.80 ± 0.30** | 90.50 ± 0.30 | **91.26 ± 0.28** | 90.74 ± 0.42 | **90.93 ± 0.14** |
| MixHop | 73.14 ± 1.62 | **75.64 ± 0.13** | **87.77 ± 0.80** | 86.12 ± 0.60 | 92.19 ± 0.26 | **92.23 ± 0.05** |
| **Basic Node Classification - Heterophilic Graphs** | | | | | |
| Datasets | Actor | | Wisconsin | | Roman-Empire | |
| Models | Bayesian | GNN-Diff | Bayesian | GNN-Diff | Bayesian | GNN-Diff |
| MLP | 37.85 ± 0.63 | **37.89 ± 0.33** | 80.16 ± 0.78 | **80.39 ± 1.07** | 66.01 ± 0.14 | **66.11 ± 0.08** |
| GCN | 30.45 ± 0.77 | **31.24 ± 0.26** | 56.47 ± 2.29 | **61.17 ± 0.78** | **50.41 ± 0.44** | 48.43 ± 0.17 |
| SAGE | 36.00 ± 0.42 | **36.11 ± 0.09** | 75.49 ± 3.95 | **76.47 ± 2.32** | 77.42 ± 0.37 | **77.65 ± 0.12** |
| APPNP | 35.08 ± 0.58 | **35.92 ± 0.21** | **83.14 ± 1.30** | 81.16 ± 1.53 | 57.33 ± 0.12 | **58.36 ± 0.34** |
| GAT | **30.30 ± 1.20** | 29.88 ± 0.23 | 52.75 ± 2.39 | **52.94 ± 0.05** | 55.29 ± 1.16 | **55.40 ± 0.04** |
| ChebNet | 37.06 ± 0.80 | **37.49 ± 0.16** | **81.56 ± 1.71** | 80.59 ± 2.83 | 79.33 ± 0.22 | **80.06 ± 0.04** |
| H2GCN | 33.63 ± 0.52 | **34.16 ± 0.21** | 78.24 ± 0.16 | **83.72 ± 0.90** | 63.70 ± 1.18 | **63.94 ± 0.07** |
| SGC | 30.14 ± 0.09 | **30.38 ± 0.17** | 56.27 ± 0.90 | **60.32 ± 1.19** | 36.96 ± 0.15 | **38.47 ± 0.16** |
| GPRGNN | 33.39 ± 2.94 | **36.84 ± 0.41** | 80.31 ± 0.84 | **81.53 ± 0.96** | 70.92 ± 0.25 | **71.76 ± 0.07** |
| MixHop | 37.32 ± 0.57 | **38.15 ± 0.38** | **81.18 ± 2.51** | 80.39 ± 0.07 | **76.68 ± 0.15** | 74.94 ± 0.13 |
| Datasets | Amazon-Ratings | | Minesweeper | | Tolokers | |
| Models | Bayesian | GNN-Diff | Bayesian | GNN-Diff | Bayesian | GNN-Diff |
| MLP | **40.41 ± 0.59** | 39.82 ± 0.20 | **80.00 ± 0.00** | **80.00 ± 0.00** | 78.13 ± 0.05 | **78.24 ± 0.05** |
| GCN | 42.91 ± 0.40 | **43.19 ± 0.14** | 80.19 ± 0.10 | **80.28 ± 0.07** | 78.54 ± 0.15 | **78.73 ± 0.08** |
| SAGE | 43.28 ± 0.99 | **43.32 ± 0.06** | 85.26 ± 0.27 | **85.78 ± 0.32** | **78.64 ± 0.11** | 78.51 ± 0.06 |
| APPNP | 37.48 ± 0.15 | **39.26 ± 0.16** | 80.04 ± 0.02 | **80.26 ± 0.02** | 78.42 ± 0.03 | **78.63 ± 0.03** |
| GAT | 44.39 ± 0.32 | **44.77 ± 0.04** | 81.75 ± 0.61 | **82.31 ± 0.06** | 79.48 ± 0.53 | **79.80 ± 0.12** |
| ChebNet | 43.79 ± 0.54 | **44.04 ± 0.18** | 86.84 ± 0.23 | **86.89 ± 0.27** | 79.10 ± 0.33 | **79.66 ± 0.19** |
| H2GCN | 42.80 ± 0.16 | **42.83 ± 0.15** | 83.32 ± 0.15 | **83.68 ± 0.32** | 78.45 ± 0.44 | **78.80 ± 0.07** |
| SGC | 38.89 ± 0.17 | **39.60 ± 0.12** | 80.26 ± 0.02 | **81.63 ± 0.24** | 78.28 ± 0.02 | **78.57 ± 0.03** |
| GPRGNN | 43.79 ± 0.73 | **44.24 ± 0.17** | 83.96 ± 0.17 | **83.97 ± 0.03** | 78.26 ± 0.11 | **78.28 ± 0.06** |
| MixHop | 42.23 ± 0.39 | **42.29 ± 0.07** | **84.28 ± 0.11** | 83.92 ± 0.12 | 79.24 ± 0.14 | **79.54 ± 0.41** |

## I.5 ANALYSIS OF THE GRAPH FEATURE ENCODER (GFE)

**Ablation Study of GFE Components**

The ablation study of GFE components aims to analyze the effectiveness of each component in Equation 2. We use 2 target GNNs, GCN and SAGE, and 2 datasets, `Cora` and `Actor`. The results are shown in Table 14. "GCN2 & GCN1 & MLP" refers to the current GFE architecture, where the three components are combined via concatenation. The same implementation applies to other architectures as well. We also include "None" for comparison, which is equivalent with p-diff, the unconditional baseline we have discussed in Table 5. Since the target GNNs only have 2 layers by our experiment design, we only consider up to 2-layer GCN (GCN2) in this analysis.

We observe that for different models on different datasets, a specific component may contribute to promising results more than other components. For example, for GCN on `Cora`, the average accuracy tends to be higher when GCN2 is included in the GFE. Similarly, GCN1 is shown to be more important for SAGE on `Cora`, while MLP may lead to better accuracy on the heterophilic graph, Actor, for both GCN and SAGE. In addition, the current GFE architecture with 3 components generally produces the best average results among all other architectures in the ablation study. We suppose this is because the current architecture incorporates all factors that may lead to promising generation outcomes and enables the automatic selection of these components during the learning process.

Furthermore, a well-designed GFE architecture, such as "GCN2 & GCN1 & MLP", enhances generation quality by improving both the average accuracy and the stability (as reflected in the standard deviation) compared to the unconditional generation model ("None").

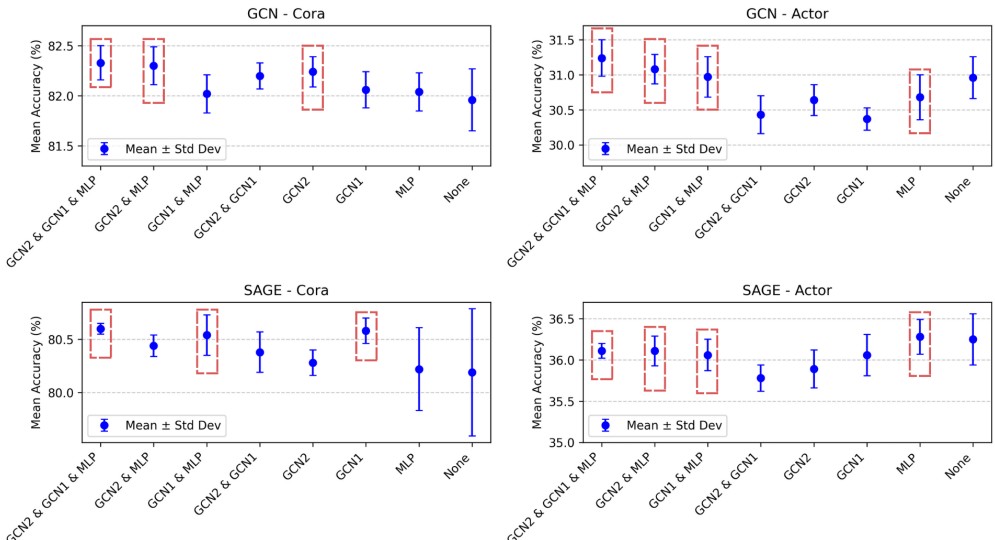

Figure 14: Results of GFE ablation study. The blue dots represent the average accuracy while the blue lines show the corresponding standard deviation. It can be observed that a specific component may lead to comparably better results than other components.

**GFE with Various Graph Convolutions**

In the previous discussion, we concentrated on the GFE architecture utilizing the GCN graph convolution. Here we extend the analysis by exploring alternative graph convolutions as the backbone of the GFE. Specifically, we investigate whether employing the graph convolution of the target models within the GFE can yield improved prediction outcomes. For instance, in the case of GAT, we compare the GFE described in Equation 2 with GFEs constructed using the GAT convolution.

We consider 3 target models, SAGE, GAT, and APPNP, and 2 datasets, `Cora` and `Actor`. For each target model, except for the GFE adopted by GNN-Diff ("GCN2 & GCN1 & MLP"), we also try GFE with 2 layers of the target model (e.g., GAT), and the concatenation with MLP (e.g., GAT & MLP). In addition, we include MLP and no graph condition as baselines for comparison.

The results in Figure 15 indicate that employing the target GNN convolution in GFE can lead to moderately higher accuracy compared to GCN-based GFE models (e.g., SAGE and GAT on `Cora`). However, this advantage is not consistently significant and may occasionally underperform (e.g., GAT on `Actor` and APPNP on `Cora`). Therefore, we conclude that the current GFE architecture utilized by GNN-Diff serves as a reasonable default for tuning any GNN. Nevertheless, exploring alternative graph convolutions is advisable if computational resources and time permit.

Additional noteworthy findings include: (1) concatenating with an MLP proves to be a versatile and effective approach that can be integrated with many graph convolutions; and (2) consistent with observations from the ablation study, parameters generated using graph conditions exhibit significantly higher stability compared to those generated without any conditions. These results further validate the efficacy of the GFE in generating parameters for GNNs.

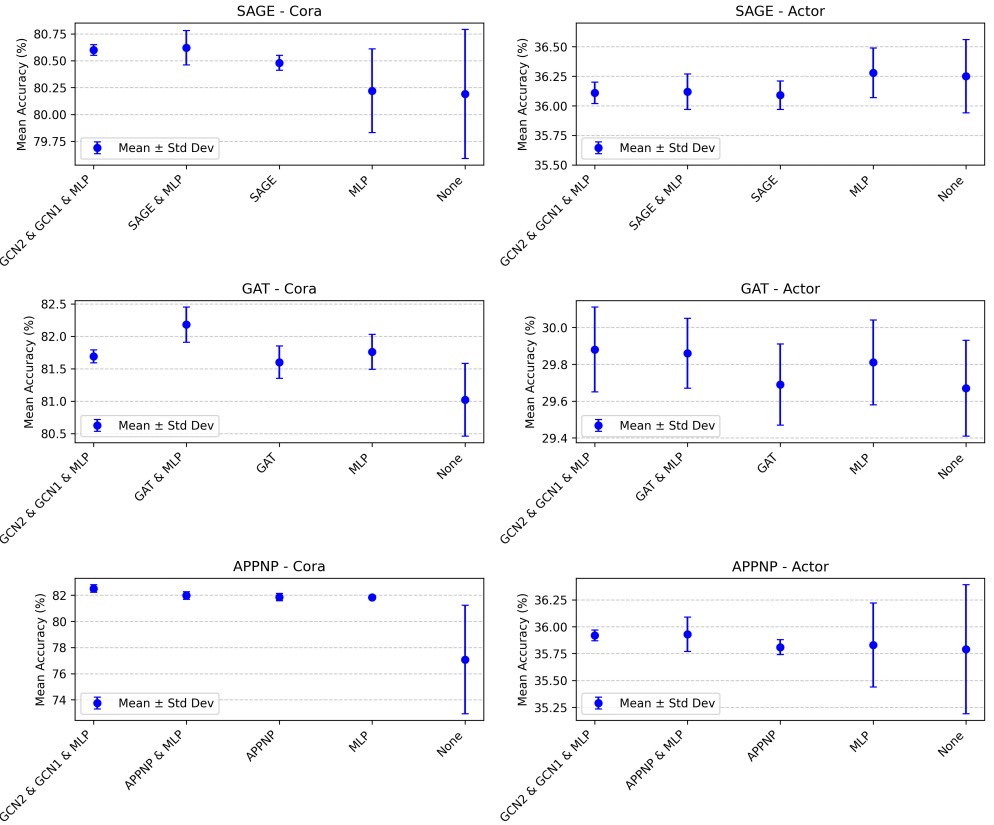

Figure 15: Results of GFEs with various graph convolutions. The blue dots represent the average accuracy while the blue lines show the corresponding standard deviation.

## I.6 GNN-Diff Time Analysis

We provide the details of the time costs of GNN-Diff in Figure 16. In general, the coarse search consumes most of the time taken by the entire GNN-Diff process. This is the inevitable cost that one shall expect for GNN-Diff to generate high-performing GNNs, though the coarse search is much more efficient compared to the baseline search methods. The parameter collection time is almost negligible for small graphs such as `Cora` and `Actor`. In contrast, it takes much longer to collect parameters for large-scale tasks, such as the node classification on `PascalVOC-SP`. This is due to the long training and validation time associated with large graphs. Similarly, the proportion of sampling time is lower for small graphs and higher for large graphs, because of the longer validation and testing time with large graphs.

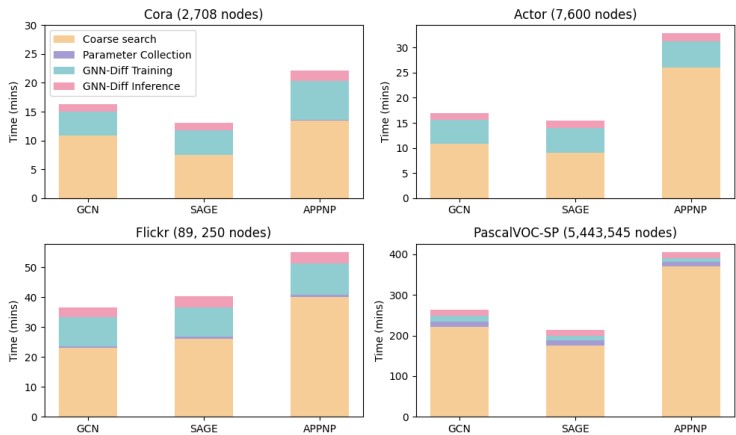

Figure 16: Time analysis of GNN-Diff. The time costs of GNN-Diff include the time for the coarse search, parameter collection, training, and inference.

## I.7 Combine GNN-Diff with Large-scale Training Algorithms

While our experiments on large graphs utilize clustering-based training for computational efficiency, future work could explore combining GNN-Diff with more advanced large-scale training algorithms.

One promising direction is integrating our method with the reversible connections proposed by Li et al. (2021), which enable the training of very deep or wide GNNs to achieve exceptional performance on large graphs. This could involve incorporating reversible connections into the architectures of target models, allowing us to evaluate whether partial generation remains effective for significantly deeper or wider GNNs. Additionally, combining GNN-Diff with the Self-Label-Enhanced training described in (Sun et al., 2021) could improve scalability and performance on large-scale datasets. Furthermore, lazy propagation in (Xue et al., 2023) offers an intriguing direction, as their efficient computation could further reduce the training costs of our framework on large graphs.

Given that GNN-Diff is a general tuning framework and these strategies are generic training algorithms, their combination could lead to a scalable, efficient, and model-agnostic approach for enhancing GNN performance on large-scale and complex graph datasets.

