# OpenReview forum: "Diffusing to the Top: Boost Graph Neural Networks with Minimal Hyperparameter Tuning"
_ICLR.cc/2025/Conference — ICLR 2025 Poster_

### Official Review · Reviewer_ti8E · 2024-10-30

**Soundness:** 3
**Presentation:** 4
**Contribution:** 3
**Rating:** 6
**Confidence:** 3

**Summary:**

This paper proposes GNN-Diff, a graph-conditioned latent diffusion model designed to enhance Graph Neural Networks (GNNs) by generating effective parameters with minimal hyperparameter tuning. This is achieved by utilizing a coarse search strategy that identifies a subset of viable hyperparameters, which then guides the model’s parameter generation. GNN-Diff claims to reduce tuning time and computational cost by creating high-performance GNN parameters through diffusion in a latent space conditioned on graph data. The authors validate GNN-Diff through extensive experiments on various tasks including node classification and link prediction with numerous model backbones, showing improved efficiency and stability.

**Strengths:**

- The idea of using the diffusion framework to help with hyperparameter tuning seems novel and reasonable.
- The experiments are thorough, spanning a diverse set of tasks and backbone models, with results that are both convincing and reproducible.
- The proposed method effectively enhances both speed and performance across most cases, showcasing its practical value.

**Weaknesses:**

- The heterophily datasets used in this paper have faced criticism for extensive duplicate nodes as noted in [1]. It would be beneficial for the authors to consider experimenting with the datasets proposed in [1] for a more robust evaluation of heterophilic cases.

- Table 5 shows that, compared to p-Diff, the improvements with GNN-Diff are relatively marginal and often inconsistent, with many results falling within one standard deviation. This brings into question the effectiveness of the GNN-Encoder design (Eq. 2).

- There is a lack of theoretical analysis.


[1] A critical look at the evaluation of GNNs under heterophily: Are we really making progress?

**Questions:**

Have the authors tried other forms of G-Encoders other than Eq. (2) and conducted ablation studies about it?

---

> ### Author Response · Authors · 2024-11-23
> **Thank you letter to Reviewer ti8E**
>
> Thank you for your detailed review and for providing constructive feedback on our work. We appreciate your assessment and acknowledge the concerns raised. Your comments have helped us identify areas for improvement, and we have added more analyses on the graph condition to our paper accordingly. Below, we provide detailed responses to your comments. We hope these clarifications will address your concerns and please feel free to ask any further questions. If possible, we kindly hope that you could consider increasing to a more favourable rating, as your support will be very helpful with the acceptance of our work. Thank you again for your time and efforts.

---

> ### Author Response · Authors · 2024-11-23
> **[Response to Weakness 1 & 2] Heterophilic datasets & clarifiacation of Table 5 results.**
>
> **[Response to Weakness 1]** Heterophilic datasets.
>
> Thank you for your valuable suggestion. We would like to clarify that 4 datasets proposed in [1] have been included in our experiments. Detailed results and analyses for these datasets are provided in **Appendix I.2**. We hope this addresses your concern, and we appreciate your attention to ensuring a robust evaluation of heterophilic cases in our study.
>
> **[Response to Weakness 2]** Clarification of Table 5 results.
>
> This is a good point. However, we would like to clarify the main benefit of the graph condition: it provides significantly more stable generation quality while achieving comparably higher average accuracy. During our experiments, we observed that while the average accuracy of GNN-Diff is not always significantly better than p-diff, its standard deviation is consistently lower. This can be attributed to the following reasons:
>
> - Similar Parameter Distributions: GNN-Diff and p-diff are trained on the same set of samples, leading them to approximate similar parameter distributions when properly trained. As a result, their average accuracies are close, although GNN-Diff generally achieves better accuracy in most cases.
>
> - Guidance from the Graph Condition: The graph condition in GNN-Diff serves as guidance to regions of promising parameters within the overall distribution. This allows GNN-Diff to generate more centered and consistently high-performing parameters. The newly added analyses in **Appendix I.5** further support this hypothesis.
>
> Thus, while the average accuracy of GNN-Diff may overlap with the variation seen in p-diff results, the primary advantage of the graph condition lies in producing dense, high-performing parameters. This contributes to the practical value of our method, especially in scenarios where reducing the impact of randomness is crucial.

---

> ### Author Response · Authors · 2024-11-23
> **[Response to Weakness 3 & Question 1] Theoretical analysis & G-Encoder ablation study.**
>
> **[Response to Weakness 3]** Theoretical analysis.
>
> Thank you for the feedback. We acknowledge the importance of theoretical analysis in advancing understanding in many areas. However, as this work is designed as an empirical study, our primary focus is on demonstrating the practical effectiveness, stability, and generalizability of GNN-Diff across diverse tasks, models, and datasets. Given that hyperparameter tuning is a practical challenge in graph neural networks, our goal was to validate the utility of GNN-Diff through extensive experiments rather than theoretical derivations. We believe that this practical focus aligns with the study's objectives and provides meaningful insights for both research and application. Nonetheless, we appreciate your suggestion and will consider including theoretical perspectives in future work.
>
> **[Response to Question 1]** G-Encoder ablation study.
>
> This is an excellent question which leads to a valuable suggestion. When designing the implementation code, we conducted comprehensive trials for the G-Encoder architecture and selected the current version as a generally promising default option for GNN-Diff. However, we recognize the importance of including such analyses in the paper to validate this choice. To address this, we have added **Appendix I.5**, which includes two analyses:
>
> - An ablation study on the components of the current G-Encoder;
>
> - Comparison of alternative graph convolutions beyond GCN and their integration with MLP as the G-Encoder.
>
> Please refer to the revised paper for further details.

---

### Official Review · Reviewer_gdji · 2024-11-01

**Soundness:** 3
**Presentation:** 3
**Contribution:** 3
**Rating:** 6
**Confidence:** 1

**Summary:**

This article introduces an interesting hyperparameter search architecture that learns graph embedding information for specific graph tasks through a designed GNN architecture. This information is then integrated into conditional latent denoising diffusion probabilistic models, using a parameter encoding module to flexibly convert the input and output dimensions of the parameters.

**Strengths:**

1. The overall expression of the article is clear and easy to understand.
2. The experiments are comprehensive, and the performance of the algorithm is impressive.
3. It connects model parameter tuning with the data itself.

**Weaknesses:**

Simply using GNNs for embedding learning on graph-structured data and adding it to the G-LDM module does not analyze why this part of the graph embedding works for the method.

**Questions:**

1. There is a spelling error at line 126 of the article, "P-Eecoder()".
2. Why use such a GNN architecture? Would using classical graph embedding frameworks like GAT or GCN be effective?
3. Clearly, graph data structures are diverse, and using just one type of GNN makes it difficult to effectively embed all types of graph data. When GNNs become ineffective, or when increasing the number of GNN layers causes the graph embeddings to become too smooth, what impact will that have on the method? Since this paper proposes a parameter tuning method, which can be considered a fundamental approach, we need to focus more on when it will fail or the lower limits of the method’s capabilities.

---

> ### Author Response · Authors · 2024-11-23
> **Thank you letter to Reviewer gdji**
>
> We sincerely thank you for your positive feedback and high evaluation of our work. We are delighted that you found our contributions valuable and appreciate your recognition of their significance. We have carefully considered your suggestions to enhance the clarity and completeness of our paper. Particularly, your suggestion on using other graph embedding frameworks such as GAT helped us to enhance the analysis on the graph condition. Below, we address your points in detail.

---

> ### Author Response · Authors · 2024-11-23
> **[Response to Weakness 1 & Question 2]  Concerns related to the graph autoencoder and graph condition.**
>
> **[Reponse to Weakness 1 & Question 2]**  Concerns related to the graph autoencoder and graph condition.
>
> Thank you for your questions. Please allow us to address *Weakness 1 and Question 2* together because they are both related to the graph autoencoder and the graph condition.
>
> **Why such a GNN architecture?**
>
> For the basic node classification, we designed the GNN architecture to effectively handle both homophilic and heterophilic graphs. Usually, graph convolution such as $\mathbf{AXW}$ aggregate neighboring nodes to integrate information in related samples and form more informative representations. However, this inevitably similarizes the neighboring node representations. This is helpful for homophilic graphs where connected nodes are generally from the same class. Since more similar representations better classify the connected nodes to the same class. However, for heterophilic graphs, where connected nodes may be from different classes, the similar representations make it hard to distinguish them to various labels. This is why we add MLP, $\mathbf{XW}$, known as the source term in relevant literature, to add back some variation in node features. The concatenation of the graph convolution and MLP is shown to be effective on both homophilic and heterophilic graphs in [1], because it allows the model to automatically decide the similarity (or smoothness) among node representations.
>
> [1]: Beyond Homophily in Graph Neural Networks.
>
> **Would GAT or GCN be effective?**
>
> The short answer is yes, compared to having no graph condition, but they may not perform as effectively as the current architecture. To validate this, we have included a comprehensive analysis of the GAE architecture in the revised paper (**Appendix I.5**). This includes an ablation study on components of the current GAE architecture and an evaluation of alternative graph convolutions such as GATConv, SAGEConv, and APPNP. The results confirm the effectiveness of the current architecture, establishing it as a reasonable default for all target GNNs. However, exploring alternative graph convolutions could also be beneficial if computational resources and time permit.

---

> ### Author Response · Authors · 2024-11-23
> **[Response to Question 1 & 3] Typo & GNN-Diff for low-capability scenarios.**
>
> **[Response to Question 1]** Typo.
>
> Thank you for pointing this out. The typo has been fixed.
>
> **[Response to Question 3]** GNN-Diff for low-capability scenarios.
>
>
> Thank you for raising the important issue of model capability—this is a good perspective we had not previously considered. It indeed represents a practical challenge when implementing GNNs. However, we would like to emphasize that our proposed method, GNN-Diff, is well-equipped to address these challenges in the following ways:
>
> - **Challenge of GNNs Becoming Ineffective**: We agree that graph data are diverse, and there is rarely a single GNN that outperforms all others across all datasets and tasks. To address this, one can conduct an efficient coarse search as an architecture search to identify promising GNNs while filtering out those that clearly underperform. The generation process can then be applied to the well-performing GNNs identified during the coarse search, further enhancing their performance.
>
> - **Challenge of Oversmoothing**: Oversmoothing is a common issue in GNNs, but our method effectively addresses it by including the number of layers as a hyperparameter. This approach is considered in our experiments on long-range graphs and can be smoothly extended to other tasks.
>
> We have also conducted experiments to evaluate whether GNN-Diff can boost the performance of ineffective GNNs. Using GCN for node classification on Cora and addressing the oversmoothing issue as an example, the results are presented in the table below. These results demonstrate that GNN-Diff not only improves accuracy but also enhances stability in terms of lowering the standard deviation, even when the target GNN is significantly affected by oversmoothing. However, the best accuracy is still achieved with an appropriate choice of the number of layers. Therefore, we recommend treating the number of layers as an important hyperparameter to tune when oversmoothing becomes a critical concern.
>
> | Number of layers | Coarse    | GNN-Diff   |
> |------------------|-----------------|------------|
> | 2-layers         | 81.89 ± 0.48    | 82.33 ± 0.17 |
> | 4-layers         | 78.10 ± 1.15    | 78.32 ± 0.14 |
> | 6-layers         | 70.02 ± 3.03    | 73.62 ± 0.24 |
> | 8-layers         | 25.61 ± 5.90    | 27.22 ± 3.57 |
> | 10-layers        | 21.30 ± 6.69    | 27.94 ± 4.74 |

---

> ### Comment · Reviewer_gdji · 2024-12-03
>
> Thank you for your feedback, which has been very helpful in addressing my concerns. However, I must admit that I am not very familiar with this field, and after reading the responses from other reviewers, I noticed several issues that I hadn’t initially anticipated. As a result, I decided to adjust my rating to a more conservative score. Nonetheless, I still believe the paper has merit and recommend it for acceptance.

---

> ### Author Response · Authors · 2024-12-03
> **Thank You Letter to Reviewer gdji**
>
> Dear Reviewer,
>
> Thank you for your engagement with our response and for the feedback you have provided throughout the review process. We sincerely appreciate the time and effort you have dedicated to evaluate our paper. While we feel deeply sorry to hear that you decrease the score, we are still very grateful for your support.
>
> If there are any following questions or concerns, we would be happy to provide further clarifications. Thank you again for your time and acknowledgment on our efforts.
>
> Sincerely,
>
> The Authors

---

### Official Review · Reviewer_TruH · 2024-11-02

**Soundness:** 2
**Presentation:** 3
**Contribution:** 2
**Rating:** 5
**Confidence:** 3

**Summary:**

In summary， this paper proposes a strategy for tuning hyper-parameters of GNNs to maximize their potential. It is achieved by collecting a set of model parameters, encoding graphs, and then learning to generate parameters for GNNs. Extensive methods have been conducted in this paper to verify the designed method.

**Strengths:**

- This paper is well-organized.
- Extensive experiments are conducted in this paper.

**Weaknesses:**

1. The technical contribution is limited given the related works.
It seems that GNN-diff is a combination of LDM and graph techniques. The sample collection method described in Section 4.1 is trivial, and the designed encoder is also widely used in existing methods. It would be better to highlight the technical contributions in comparison with the different methods mentioned in the related work section.

2. The experiments seem somewhat unrigorous and fail to adequately highlight the efficiency and effectiveness of GNN-diff.
- Baseline selection: While numerous target GNNs and datasets are utilized, the paper employs only a few comparable baselines. It appears that methods from "Advanced Methods for Hyperparameter Tuning" should be incorporated to validate the designed parameter tuning strategy more effectively.
-  Evaluations of the effectiveness of Coarse search: In Tables 1-4, the "Coarse" variant struggles to outperform the "Random" and "Grid" baselines. With these results, justifying the effectiveness of the coarse search approach is challenging.

3. The transferability of GNN-Diff is limited due to its hyper-parameter tuning being tailored for a specific dataset, relying on parameter collection and the GAE model.

**Questions:**

The upper bound of the proposed method remains unclear. It is uncertain how a well fine-tuned GCN compares to lightly fine-tuned, more expressive models.

---

> ### Author Response · Authors · 2024-11-23
> **Thank you letter to Reviewer TruH**
>
> Thank you very much for the time and effort you dedicated to reviewing our paper, as well as for your valuable suggestions. We especially appreciate your insights on strengthening the technical contributions of our work. In the revised paper, we have added additional experiments and discussions to further clarify and enhance our work. Here, we would like to take this opportunity to address your concerns and clarify some potential misunderstandings. If possible, we sincerely hope that you could consider increasing the rating. Again, we greatly appreciate your expertise and support in the review process.

---

> ### Author Response · Authors · 2024-11-23
> **[Response to Weakness 1 & 2] Technical contribution and supplementary experiments.**
>
> **[Response to Weakness 1]** Technical contribution.
>
> Thank you for raising this concern. In this response, we will first highlight the novelty and contribution of our method, and then discuss the modifications we have made in the revised paper to better improve the technical contribution.
>
> To start with, we would like to clarify that the main contribution of our work is not to propose a new model component that has never been adopted before in relevant literature; but to explore an efficient and novel training strategy for GNNs with the minimal hyperparameter tuning, which has not been sufficiently studied so far. We agree with your concern that LDM and the GAE architecture are not new, and we have cited relevant works in our paper. However, lowering the hyperparameter tuning costs for GNNs with a parameter generative model is a brand new area for GNN research. The relationship between model components and our novelty is just like the ingredients and the recipe. The ingredients such as LDM, the graph convolution in GAE, and the sample collection are easy to obtain, but the design of the recipe (i.e., what ingredients to include and how to combine them for effectiveness and efficiency) is the true value of our work.
>
> Then, to better improve the technical contribution of our work, we carefully considered the suggestion of more baselines for comparison. We will discuss more details in the response to Weakness 2.
>
> **[Response to Weakness 2-1]** 120 supplementary experiments with Bayesian optimization as a baseline.
>
> Thank you for the suggestion. We have revised the paper accordingly. Here are a few key points that we would like to highlight:
>
> - In "Advanced Methods for Hyperparameter Tuning", we mentioned 3 types of techniques: parameter-free optimization, Bayesian optimization, and Hyperband. For Hyperband, we have used one of its variants, the coarse-to-fine (C2F) search. So, our revision majorly focused on parameter-free optimization and Bayesian optimization.
>
> - Parameter-free optimization: While parameter-free optimization is a useful tool for tuning hyperparameters related to neural network training, such as the learning rate, this method is incapable of handling model hyperparameters such as the ChebNet K and APPNP alpha. So it is not considered as our baseline. We further clarify this limitation in Appendix A - Advanced Methods for Hyperparameter Tuning.
>
> - Bayesian optimization: Bayesian optimization was not chosen as our comparison baseline because it is not common to apply Bayesian optimization for GNN tuning in GNN-related research. However, it indeed can be applied. Given that our paper is related to hyperparameter tuning, we do find your suggestion very valuable, and we have added experiments of Bayesian optimization. Please refer to **Appendix I.4** in the revised paper for more details.
>
> **[Response to Weakness 2-2]** Clarification on the coarse search results.
>
> We suppose here is some misunderstanding. We will provide an explanation on why this is not an issue - and maybe even the "advantage'' of our method. But please kindly advise if our explanation does not address your question :)
>
> Firstly, we agree with your observation that the coarse search underperforms compared to random and grid search in most cases. This is expected, as the coarse search operates over a very limited search space, which sacrifices exploration for tuning efficiency. However, our method, GNN-Diff, uses the coarse search as an initial step, then improves upon it through parameter generation.
>
> Based on our experimental results, we make two key observations:
>
> - GNN-Diff is able to outperform both random and grid search, even when starting from a relatively weak point provided by the coarse search;
>
> - GNN-Diff requires significantly less time than both random and grid search.
>
> These findings together demonstrate that GNN-Diff is an effective method for enhancing GNN performance within a short time frame. The coarse search provides a reasonable starting point, and GNN-Diff is able to build upon that to achieve superior results efficiently.

---

> ### Author Response · Authors · 2024-11-23
> **[Response to Weakness 3 & Question 1] Transferability & upper bound of GNN-Diff.**
>
> **[Response to Weakness 3]** Transferibility and potential extension.
>
> Thank you for bringing the transferability to our attention. We understand your concern as the transferability is indeed not in the scope of our work. Yet, we would like to emphasize that our method is designed for the scenario in which one would like to boost GNN performance for a specific task to achieve the top potential. If transferability is taken into account, it may lead to a sacrifice of accuracy. In addition, the current design of GNN-Diff shows strong ability to boost various GNN models across diverse tasks and datasets. Thus, we expect that, with some adjustments, it is possible for it to gain transferability. We plan to explore further generalizations of our approach in future work, which may involve the method proposed in [1]. We appreciate your insights, and we will take them into consideration for future improvements.
>
> [1] Diffusion-based Neural Network Weights Generation.
>
> **[Response to Question 1]** Clarification on the upper bound of GNN-Diff.
>
> Thank you - this is a good question. The global upper bound of our method for a specific task or dataset is indeed hard to measure, but we would like to clarify two key points regarding the intention of our method.
>
> Firstly, our method is designed to be model-agnostic. The primary goal is to enhance the performance of any given GNN model, allowing it to reach its maximum potential. As such, the upper bound is not tied to any particular GNN but to how well our method can help fine-tune the performance of various models across different tasks and datasets.
>
> Secondly, it is important to note that different GNNs may be better suited for different types of tasks or datasets. Some models may excel in certain scenarios, but underperform in different contexts. So, a more expressive model may not necessarily exist for all graph tasks. Classic GNNs, such as GCN, may even serve as strong baselines as shown in [2]. Therefore, rather than focusing on determining which GNN is inherently the most powerful, our method aims to optimize the performance of the target GNN on a specific task, helping it achieve its best possible results.
>
> [2] Classic GNNs are Strong Baselines: Reassessing GNNs for Node Classification

---

> ### Author Response · Authors · 2024-11-26
> **Follow-up Letter from the Authors**
>
> Dear Reviewer,
>
> Thank you for your thoughtful and constructive feedback on our paper. We greatly appreciate the time and effort you have dedicated to reviewing our work and have carefully addressed your comments and suggestions.
>
> To facilitate your review of our explanations and revisions, we have summarized our previous response below:
>
> - We have conducted 120 additional experiments to further strengthen the technical contributions of our work. These include **a detailed comparison of GNN-Diff with Bayesian optimization implemented using Optuna (Appendix I.4, page 32)**.
>
> - For other concerns, such as the underperformance of coarse search, we have provided clarifications to demonstrate how it reflects the effectiveness of GNN-Diff. However, further suggestions are always welcome if our current explanation does not meet your expectations.
>
> If you have any further comments or suggestions, we would be happy to address them promptly. Thank you again for your valuable feedback and thoughtful engagement with our work.
>
> Sincerely,
>
> The Authors

---

### Official Review · Reviewer_8AwD · 2024-11-03

**Soundness:** 2
**Presentation:** 3
**Contribution:** 3
**Rating:** 6
**Confidence:** 4

**Summary:**

The paper introduces GNN-Diff, a latent diffusion model designed to enhance GNN training while minimizing the need for hyperparameter tuning. The author claim that this approach not only improves GNN performance but also demonstrates significant stability across various tasks.

**Strengths:**

1.	The overall motivation is clear: improving hyperparameter search strategies can save time and enhance performance. This is an area that has not been explored too much in GNN training so far.

2.	The presentation is clear, with well-organized results and analysis provided for better interpretability.

**Weaknesses:**

1.	The reported performance is a significant concern, as many results are noticeably lower than expected. For instance, nearly all results in Table 2 fall substantially short of those reported in existing literature and on the OGBN leaderboard: for example, I believe that GCN on OGBN-Arxiv can achieve 71.6% (as opposed to the reported 69.2%), APPNP can reache 71.8% (versus the reported 55.05%), and GraphSAGE can easily achieve around 79% on OGBN-Products, compared to the reported 75.6%. This discrepancy raises questions about whether the authors adhered to standard settings, whether the reported accuracy is reliable, and, if not, how the experimental setup was configured. Clearer details on these aspects would help clarify the validity of the reported results.

2.	The motivation for certain components is unclear. For instance, why do the authors choose to incorporate GAE in the training process?

3.	An ablation study illustrating the impact of the proposed techniques would strengthen the findings.

4.	The authors use classical GNNs for downstream tasks, but employing SOTA GNNs are necessary. For example, RevGAT[1], SAGN[2] and LazyGNN[3]. Many advanced architectures perform well without extensive parameter tuning, and using these models could better showcase the contributions of this work.

5.	Figure 7 reports only running time, without any indication of memory cost. Additionally, if three different models are trained in this work, I am wondering how the runtime remains lower than that of a random search? A comparison with coarse search would also be useful.

6.	Can the proposed techniques be applied to graph classification?

[1] Training Graph Neural Networks with 1000 Layers

[2] Scalable and Adaptive Graph Neural Networks with Self-Label-Enhanced training

[3] LazyGNN: Large-Scale Graph Neural Networks via Lazy Propagation

**Questions:**

1. Could you explain the reason of the claim that concatenating different receptive fields in message passing can handle both homophilic and heterophilic graphs?

---

> ### Author Response · Authors · 2024-11-22
> **Thank you letter to Reviwer 8AwD**
>
> Thank you for your thoughtful review and valuable feedback. Based on your comments, we recognize your expertise in large-scale graphs and appreciate your insights. We have revised our paper to address your concerns, clarified potential misunderstandings, and made improvements as detailed below. We hope these revisions demonstrate the soundness of our work. We also hope you could kindly reconsider the rating. Thank you again for all the constructive suggestions and your time.

---

> ### Author Response · Authors · 2024-11-23
> **[Response to Weakness 1] Factors lead to the gap in the experiment results.**
>
> Thank you for highlighting this point. We also had the same observation during our experiments. As discussed in Section 5.1 - Reproducibility and comparability, this discrepancy is primarily due to differences in data split and model architectures. We would like to emphasize that our experimental settings mostly align with standard practices in the original paper of OGBN datasets [1], except for the following aspects:
>
> - Cluster vs. Full training: As mentioned in Section 5.1 - Experiment details, we conducted all experiments on NVIDIA 4090 GPU with 24GB memory. Given the limited memory, the large graphs are split into clusters following [2]. This may not be necessary for relatively smaller datasets such as Flickr, but we apply the same setting to maintain consistency. Clusters sometimes lead to inferior performance especially comparing with the full graph training adopted by most Leaderboard GNNs. However, this does not affect the validity of experiments to show GNN-Diff's ability in boosting GNN performance on large graphs.
>
> - No tricks vs. Tricks applied: We didn't apply tricks such as skip connection, which is considered in GraphSAGE [1]. The inititial goal of our experiments is to investigate the top potential of vanilla GNNs.
>
> - 2-layer GNN vs. 3 or more layers: We adopted 2-layer GNNs in large graph experiments following the same setting as the basic node classification experiments. In comparison, GNNs in [1] adopt 2-3 layers, while GNNs in [3] adopt 2-10 layers. This factor may not be as influential as the previous two, but we suppose it may still cause some differences in accuracy.
>
> To further validate the effectiveness of our method, we repeat the experiment for GCN and SAGE on OGB-arXiv strictly following the settings in [1]. In the table below, we present the result, and it is shown that GNN-Diff can achieve better accuracy with lower standard deviation than the result reported in [1] (GCN 71.74\% $\pm$ 0.29\% and SAGE 71.49\% $\pm$ 0.27\%).
>
> | Method-Dataset          |      Grid       | Random     | Coarse     | C2F        | GNN-Diff   |
> |-----------------|------------|------------|------------|------------|------------|
> | **GCN-arXiv**   | 71.90 ± 0.24 | 71.39 ± 0.27 | 71.38 ± 0.26 | 71.74 ± 0.27 | 72.04 ± 0.18 |
> | **SAGE-arXiv**  | 71.83 ± 0.24 | 71.42 ± 0.33 | 71.12 ± 0.21 | 71.46 ± 0.19 | 71.96 ± 0.12 |
>
> [1] Open Graph Benchmark: Datasets for Machine Learning on Graphs
>
> [2] Cluster-GCN: An Efficient Algorithm for Training Deep and Large Graph Convolutional Networks
>
> [3] Classic GNNs are Strong Baselines: Reassessing GNNs for Node Classification

---

> ### Author Response · Authors · 2024-11-23
> **[Response to Weakness 2, 3, & 4] Motivation of the graph autoencoder (GAE), ablation study, and SOTA GNNs.**
>
> **[Response to Weakness 2]** Motivation of GAE.
>
> Thank you for the question; it’s an excellent point. We previously put the motivation of GAE in Section 5.3 - Ablation Study on the Graph Condition. However, we recognize that placing it in Section 4.2 - Graph Autoencoder, where GAE is formally introduced, will make it clearer for readers. We have made this adjustment accordingly. Please see **line 194-197** in the revised paper.
>
> To answer the question, the motivation for including GAE is threefold. (1) Contribution: The role of specific data characteristics in parameter generation still remains underexplored. To fill this gap, GNN-Diff leverages the graph data and structural information as the generative condition. (2) Usage: GAE is used to construct the graph condition, which is considered as a data and graph structure-aware guidance on the GNN parameter generation. (3) Accuracy \& Stability: We show with the ablation study in Table 5 that GNN parameters generated with the graph condition generally lead to better average accuracy and higher stability across various tasks.
>
> Further, in terms of the training process, we train the three components one by one in a consecutive training flow. The training of GAE is prior to the training of G-LDM, because its encoder will be used to construct one of the inputs of G-LDM, that is, the graph condition.
>
> **[Response to Weakness 3]** Ablation Study.
>
> We appreciate this suggestion and have enriched the ablation study accordingly. Please refer to **Appendix I.5** for detailed experiment results and analyses.
>
> **[Response to Weakness 4]** SOTA GNNs.
>
> Thank you for highlighting three very intriguing papers on large-scale graphs. As a matter of fact, we originally considered such works during our experiment design, but we eventually decided to focus on exploring "graph convolution architectures'' rather than "efficient GNN training algorithms'' (i.e., these papers share the same characteristic in emphasizing efficient training strategies rather than proposing new convolutions). Nevertheless, we appreciate your reminder of their relevance, and we acknowledge their importance for scalability in real-world challenges. To address this, we have cited these papers and added a discussion on the potential of combining our method with these algorithms. Please find the discussion in **Appendix I.7** of the revised paper.
>
> To further clarify, please see below for more explanations of why the three papers all focus on efficient training algorithms rather than proposing new architectures:
>
> - [Training Graph Neural Networks with 1000 Layers] This paper proposes reversible connections to enable the training of very deep or very wide GNNs for large graphs. The proposed algorithm is generic and can be theoretically applied to any GNNs, including GCN, SAGE, GAT, etc.
>
> - [Scalable and Adaptive Graph Neural Networks with Self-Label-Enhanced Training] SAGN proposed in this paper gathers multiple-hop representations with the attention mechanism. This is very similar to one of our target model, MixHop. Since MixHop is not included in the large graph experiments due to its computational complexity, this paper can be a very good direction for our future work. This paper also proposes Self-Label-Enhanced training, which is also a generic training algorithm.
>
> - [LazyGNN: Large-Scale Graph Neural Networks via Lazy Propagation] This paper introduces lazy propagation and combines it with generic sampling strategies. Lazy propagation can be regarded as a way to organize and compute GNN layers, and the graph convolutional layers of LazyGNN are very similar to one of our target model, APPNP.
>
> Accordingly, we did not include these works as target models in our main experiments but have discussed GNN-Diff's extension to SOTA training algorithms in **Appendix I.7**.

---

> ### Author Response · Authors · 2024-11-23
> **[Response to Weakness 5 & 6, and Question 1] Memory & time, graph classification, and explanation of G-Encoder architecture.**
>
> **[Response to Weakness 5]** Memory and time concerns.
>
> Thank you for your questions and suggestions. Below, we address them point by point:
>
> - **Memory Costs**: Memory costs were not reported as they were not a significant issue in our experiments. All experiments were conducted on a single NVIDIA 4090 GPU with 24GB memory, which was sufficient for all tasks. For large datasets, such as large graphs and long-range graphs, we employed clustering and batch training to mitigate memory usage. Most experiments required far less than the available memory.
>
> - **Why GNN-Diff is Faster Than Random Search**: GNN-Diff outperforms random search in efficiency for two main reasons:
>
>   - **Reduced Search Space**: GNN-Diff uses coarse search, which has a search space only half the size of the random search.
>
>   - **Minimal Training Overhead**: Although GNN-Diff involves three trainable components, their training time is very short compared to the coarse search. The GAE training is equivalent to one run of GNN training, while search methods involve many more runs depending on the search space. The PAE and G-LDM training are computationally efficient due to the partial generation technique in our implementation. This approach considers only the last layer parameters, significantly reducing computational costs. Additionally, PAE encodes parameters into a low-dimensional space, which enhances the training and inference efficiency of G-LDM.
>
> - **Comparison with Coarse Search**: Following your suggestion, we have updated **Figure 7** to include the coarse search time. The revised figure clearly demonstrates that the coarse search accounts for the majority of GNN-Diff's time, supporting our claim that the training and inference time of GNN-Diff is relatively short.
>
> We hope these clarifications could address your concerns.
>
> **[Response to Weakness 6]** Graph classification.
>
> Yes, our method can be adapted to graph classification. In most cases, the major difference between graph-level and node-level classification tasks is the graph pooling process. So, we may extend to graph classification by adding a pooling layer to the GAE architecture and also the target models. The pooling layer obtains global graph representations for downstream tasks. In **Appendix I.3**, we discuss more details related to graph classification along with the experiment results on Fake News datasets [4] to show the effectiveness of our method in graph classification.
>
> [4] User Preference-aware Fake News Detection
>
> **[Response to Question 1]** Explanation of G-Encoder architecture.
>
> Thank you for the question, and here is the explanation. As we know, graph convolutions such as $\mathbf A\mathbf X\mathbf W$ and $\mathbf A^2\mathbf X\mathbf W$ aggregate neighboring node features to construct more informative representations, which inevitably similarizes neighboring node representations. This is useful for homophilic graphs, where connected nodes are usually from the same class. As similar representations better map the connected nodes to the same class. However, this may be harmful for heterophilic graphs, where connected nodes are usually from different classes. This is why some GNNs may perform even worse than MLP on heterophilic graphs.
>
> GNNs that can handle both homophilic and heterophilic graphs well typically incorporate dynamics that can bring the variations of features back, e.g., through adding the source terms in a similar way as skip connection [5, 6]. In this way, the model can choose to smoothen or differentiate node representations in the learning process. Accordingly, we adopt equation (2) with the concatenation of different receptive fields (e.g., graph convolutions $\mathbf A^2\mathbf X\mathbf W_1,\mathbf A\mathbf X\mathbf W_2$, and the source term $\mathbf X\mathbf W_3$). Furthermore, we note that our message passing structure (e.g., equation (2)) is also aligned with some recently developed GNNs, such as H2GCN [7], which serves as one of the state-of-the-art models in terms of learning with both homophilic and heterophilic graphs.
>
>
> [5] From Continuous Dynamics to Graph Neural Networks: Neural Diffusion and Beyond.
>
> [6] GREAD: Graph Neural Reaction-Diffusion Networks.
>
> [7] Beyond Homophily in Graph Neural Networks.

---

> ### Author Response · Authors · 2024-11-26
> **Follow-up Letter from the Authors**
>
> Dear Reviewer,
>
> Thank you for your thoughtful and constructive feedback on our paper. We sincerely appreciate the time and effort you have devoted to reviewing our work and have carefully considered your comments and suggestions.
>
> To assist in reviewing our explanations and revisions, we have provided a brief summary of our previous response below:
>
> - We have addressed several of your concerns through revisions to the paper and additional experiments, particularly focusing on **the motivation for GAE (Section 4.2, page 4), the ablation study (Appendix I.5, pages 33-34), and graph classification (Appendix I.3, pages 30-31)**. These updates aim to improve the clarity and contributions of our work, and we hope they align with your expectations.
>
> - Some of your other recommendations, such as including additional SOTA GNNs on large graphs, extend beyond the immediate scope of this study. However, we have included discussions on these directions as potential future work in our revised submission (**Appendix I.7, page 35**) to illustrate how our approach could be further expanded.
>
> If you have any additional comments or suggestions, we would be happy to address them promptly. Thank you again for your valuable feedback and engagement with our work.
>
> Sincerely,
>
> The Authors

---

> > ### Comment · Reviewer_8AwD · 2024-11-28
> >
> > Thank you for your response. Most of my concerns have been addressed, and I have increased my score accordingly. However, a few issues remain unresolved:
> >
> > 1. In the response to Weakness 1, the author states that GPU memory limitations necessitated the use of subgraph sampling in the original experiments. However, the performance table provided in the response appears to report accurate performance metrics without memory concerns. Based on my experience, a GPU with 24GB of memory is sufficient for most datasets used, including ogbn-products. Therefore, I am still unclear why the author chose to report significantly lower performance in the original submission.
> >
> > 2. In the response to Weakness 5, the author mentions that memory cost is not a significant issue for the experiments.
> > This statement seems contradictory to the explanation provided in Weakness 1, where GPU memory limitations were cited as a reason for using subgraph sampling.
> >
> > 3. Regarding the question, based on the response, it appears that the author employed a specialized GNN that concatenates different receptive fields and simply adds a linear classifier on top for supervised learning. Since GAE is designed for self-supervised learning with a reconstruction objective, I think the proposed method does not qualify as a GAE. I recommend that the author avoid using the term GAE to prevent misunderstandings.
> >
> > Besides, i am wondering if the author could provide an ablation study regarding the number of receptive fields in that GNN with of performance on heterophilic graphs to see the reason why the author chooses to include it from 0 hop to 2 hops.
> >
> > I hope these remaining concerns can be addressed to further improve the quality of the work.

---

> > > ### Author Response · Authors · 2024-11-28
> > > **Response to Unresolved Issues**
> > >
> > > We sincerely appreciate the time and effort you have invested in reviewing our work and providing valuable feedback. Below we address the unresolved issues one-by-one.
> > >
> > > **[Response to Unresolved Issue Point 1]** Memory limitaion & further explanations
> > >
> > > Thank you for raising this concern. We deeply appreciate the opportunity to clarify and address your question. Below are some key points:
> > >
> > > - **Limited computational resources**: While our proposed method is not a big concern for a single 4090 GPU, full-graph training can be time-consuming for baseline methods like grid search. To address this challenge and maintain consistency, we used cluster training for time efficiency and applied it across all large graph datasets.
> > >
> > > - **Out-of-memory (OOM) issue for OGBN-Products**: We revisited the OGBN-Products dataset on our 4090 GPU and unfortunately encountered OOM errors again. While we cannot fully explain this behavior, we are happy to share screenshots or additional evidence if needed.
> > >
> > > - **Supplementary experiments**: To complete the supplementary experiments within the rebuttal period, we rented multiple GPUs online. This allowed us to execute these experiments more quickly compared to those conducted on our own device.
> > >
> > > We hope these clarifications address your concerns. If additional details or evidence are required, we would be more than willing to provide them. Moreover, we would like to emphasize that, despite some resource limitations, our experiments effectively validate the proposed method, even if they do not achieve the same results as full-training benchmarks.
> > >
> > > **[Response to Unresolved Issue Point 2]** Memory is a concern, or not?
> > >
> > > We apologize for any confusion caused by our previous response, and we sincerely thank you for pointing this out. To clarify, when we mentioned that "memory cost is not a significant concern," we were specifically referring to the experiments presented in our paper, where cluster training was employed. For full-graph training, however, memory limitations remain a challenge.
> > >
> > > Please let us know if any confusion persists, and we will be happy to provide further clarification.
> > >
> > > **[Response to Unresolved Issue Point 3]** Name of GAE
> > >
> > > Thank you for the thoughtful suggestion. After careful consideration, we have decided to rename "Graph Autoencoder (GAE)" to "Graph Feature Encoder (GFE)." This new name better reflects the purpose of the module, which is to encode information from both the graph structure and node/graph features. We have updated the paper accordingly, with all changes highlighted in red for clarity.
> > >
> > > **[Response to Unresolved Issue - Ablation]**
> > >
> > > Thank you for your question. Based on our understanding, you are requesting an ablation study on the number of hops or the number of graph convolutions in the GFE-encoder. Please let us know if we have misunderstood your request.
> > >
> > > We would like to clarify that such an ablation study is included in **Appendix I.5 - Ablation Study of GFE Components (page 33)**. In this section, we compare the GFE with no graph convolution (MLP), 1-hop (MLP \& GCN1), and 2-hop (MLP \& GCN2). Additionally, we provide other combinations, such as solely GCN1 or GCN2, and GCN1 \& GCN2. This analysis is conducted using both homophilic (Cora) and heterophilic (Actor) graphs.
> > >
> > > If you would like us to include more ablation studies, we are happy to do so. However, due to the limited time available for paper revision, we may only be able to provide additional results via anonymous GitHub links initially. These results could then be incorporated into the final camera-ready version if the paper is accepted. Please advise if further ablation analyses are expected.
> > >
> > > **[A Sincere Request for Further Increasing the Score]**
> > >
> > > If you feel that our responses and the improvements to the paper sufficiently address your concerns, we would be truly grateful if you could provide us with a more favorable score. Your support and acknowledgment would mean a lot to us and would further encourage our efforts in advancing this research direction.
> > >
> > > Thank you again for your thoughtful review and consideration.

---

> > > > ### Comment · Reviewer_8AwD · 2024-12-01
> > > >
> > > > Thank you for your response. I think the advantages outweigh the disadvantages given comprehensive results provided, so I have increased my score to a positive value. However, I recommend revising the experiments section to use the commonly reported values from other baselines.
> > > >
> > > > Additionally, I found the results of the ablation study in Appendix I.5 interesting and have some questions:
> > > >
> > > > 1. Did you experiment with more than two GCN layers (hops) on the current or other heterophilic datasets?
> > > > Does including additional hops introduce issues such as over-smoothing?
> > > >
> > > > 2. Why does SAGE perform best on heterophilic graphs when using an MLP? Since SAGE is quite similar to GCN except for the pooling mechanism, if the MLP performs best, does that imply that incorporating neighbor information is not beneficial for certain heterophilic graphs? I find this intriguing and believe it would be valuable to provide more results on this topic.
> > > >
> > > > 3. Did you consider or experiment with other strategies for heterophilic graphs instead of skip connection/concatenation?

---

> > > > > ### Author Response · Authors · 2024-12-03
> > > > > **Follow-up Response to Ablation Study-related Questions**
> > > > >
> > > > > Thank you for your thoughtful questions and support. Please find our detailed responses below. Additionally, we have conducted supplementary experiments to further support our discussions. Please find all supplementary materials in https://anonymous.4open.science/r/Follow_up_responses-B7C8/README.md.
> > > > >
> > > > > **[Response to Revision of Large Graph Experiment Results]**
> > > > >
> > > > > Thank you for your valuable suggestion. We are actively working on the revision. As previously mentioned, performing a grid search with full training requires significant time, and we may not be able to finalize the results and the revised paper before the discussion deadline. We sincerely hope for your understanding and appreciate your patience.
> > > > >
> > > > > **[Response to Ablation Study - Question 1]** More hops
> > > > >
> > > > > This is indeed an intriguing aspect to explore. To build on our previous analysis, where we experimented with GFE using "GCN1" and "GCN2", we conducted additional experiments by increasing the number of GCN layers from 3 to 5. The results, presented in **Supplementary Material 1**, indicate that the number of GCN layers has minimal impact on generative quality and may even improve model accuracy.
> > > > >
> > > > > We hypothesize that over-smoothing is less of a concern for GFE, as its primary role is to extract and provide graph information. As long as the graph features and structural information are effectively captured, over-smoothing appears to be mitigated. Furthermore, the inclusion of an MLP enhances the architecture, enabling the model to dynamically adjust the feature variations within the graph condition, thereby further alleviating potential over-smoothing issues.
> > > > >
> > > > > **[Response to Ablation Study - Question 2]** SAGE
> > > > >
> > > > > This is an excellent observation. Does this suggest that incorporating neighbor information may not always be beneficial for certain heterophilic graphs? In short, yes. As discussed, smoothing node features can sometimes be detrimental for heterophilic graphs. In such cases, even a single linear layer (MLP) can outperform traditional GNNs like SAGE, GCN, and GAT.
> > > > > To support this statement, we have included experimental results from [1] in **Supplementary Material 2**. These findings indicate that this behavior is typically observed when the heterophilic level of a graph is particularly high.
> > > > >
> > > > > **In Supplementary Material 3**, we have included an additional experiment: SAGE-Actor with MLP2 as the GFE encoder, where MLP2 consists of two linear layers. The results indicate that the generative performance improves compared to using a single-layer MLP. This suggests that feature representation, rather than graph structure, plays a more critical role in this specific task.
> > > > >
> > > > > The tricky part of this problem is - we don't really know (or it takes quite a lot efforts to measure) whether a graph is very heterophilic or not. For this reason, we designed the GFE to be generic, ensuring it performs effectively on both homophilic and heterophilic graphs in practice. While this decision prioritizes better generalization, it may lead to sub-optimal performance in specific cases like SAGE-Actor. Nevertheless, the architecture still achieves promising results.
> > > > >
> > > > > [1] Understanding convolution on graphs via energies
> > > > >
> > > > > **[Response to Ablation Study - Question 3]** Other strategies for heterophilic graphs
> > > > >
> > > > > While including the source term (e.g., skip connection or concatenation) is the most commonly adopted approach in the relevant literature, other strategies exist as well. For instance, the heterophily level of graphs can be modified through adjacency re-weighting or re-wiring methods [2, 3]. These approaches address the issue of inter-class connections by directly altering the adjacency structure, such as reducing edge weights or removing connections between nodes of different classes.
> > > > >
> > > > > In our work, we chose not to adopt this strategy as we aimed to preserve the original adjacency matrices provided with the datasets. Nevertheless, we emphasize that the GFE architecture is highly flexible and can naturally extend to incorporate such strategies for handling heterophilic graphs, offering opportunities for future exploration.
> > > > >
> > > > > [2]: Neural Sheaf Diffusion: A Topological Perspective on Heterophily and Oversmoothing in GNNs.
> > > > >
> > > > > [3]: CurvDrop: A Ricci Curvature-Based Approach to Prevent Graph Neural Networks from Over-Smoothing and Over-Squashing.

---

> ### Author Response · Authors · 2024-11-28
> **Thank You for Valuable Further Suggestions**
>
> Dear Reviewer,
>
> We would like to extend our warmest thanks to you for taking the time to review our responses and for increasing our score. We deeply appreciate your constructive suggestions and insights, which will undoubtedly contribute to the further improvement of our work.
>
> We are committed to addressing the concerns you have raised and will follow up with our detailed responses as soon as possible. We believe it won't take long and thanks in advance for your patience.
>
> Best regards,
>
> The Authors

---

> ### Author Response · Authors · 2024-12-01
> **Thank You for Increasing the Score :)**
>
> Dear Reviewer,
>
> Thank you very much for further increasing the score of our submission and for your thoughtful questions! We also find these questions very interesting and inspiring to explore. While we are unable to revise the paper at this stage, we are committed to investigating these points and will share any notable findings through an anonymous GitHub link (the updated findings can be included in the camera-ready version of the paper if it is accepted). We will follow up as soon as possible.
>
> Thank you once again for your valuable feedback and support.
>
> Warm regards,
> The Authors

---

### Author Response · Authors · 2024-11-22
**Response to Area Chair and All reviewers**

We would like to express our sincere gratitude to the area chair and all reviewers for their efforts and time in evaluating our paper. Your constructive comments and suggestions have been invaluable in improving the quality and clarity of our work.

Our paper introduces GNN-Diff, a generative diffusion framework designed to boost the performance of GNNs through efficient hyperparameter tuning. Through extensive experiments, we demonstrate that GNN-Diff consistently enhances GNN performance, achieves high stability, and generalizes effectively to unseen data.

In response to the reviewers’ insightful comments, we have made revisions to the paper. Some major changes include:

- Extension to graph classification tasks with pilot experiment results (Appendix I.3, pages 30-31)

- Comparison with Bayesian optimization in 120 experiments (Appendix I.4, page 32)

- Further ablation and analyses on GAE to show the effectiveness of the current architecture and explore other potential architectures (Appendix I.5, pages 33-34)

We hope these updates address the concerns raised and further highlight the significance of our contributions. We sincerely hope the reviewers will reconsider increasing their ratings in light of these revisions, as we believe this work provides a meaningful advancement in making GNNs more efficient and robust for real-world applications. Your support would be greatly appreciated in promoting this work to the broader research community.

Thank you again for your efforts and consideration.

---

### Comment · Area_Chair_Pfap · 2024-11-28

I would like to encourage the reviewers to engage with the author's replies if they have not already done so. At the very least, please
acknowledge that you have read the rebuttal.

---

### Author Response · Authors · 2024-11-28
**Clarification on "GAE" and "GFE"**

Dear Area Chair and Reviewers,

The name "Graph Autoencoder (GAE)" has been updated to "Graph Feature Encoder (GFE)" in the latest version of the revised paper, following the suggestion of Reviewer 8AwD. This change aims to better distinguish the module from traditional Graph Autoencoders that use reconstruction loss. However, please feel free to use "GAE" and "GFE" interchangeably in the discussion.

Best regards,

The Authors

---

### Author Response · Authors · 2024-12-03
**A Sincere Request to Reviewers for Kindly Increasing Your Score**

Dear Reviewers 8AwD, TruH, and ti8E,

We are writing to kindly seek your continued support for the acceptance of our work. As you may have noticed, one reviewer has recently decreased the score and confidence due to limited familiarity with this research area, though the merit of our work is still acknowledged. While we greatly respect the opinions and decisions of all reviewers, we feel deeply sorry to see this situation, as it may have an unintended adverse impact on the final decision.

**We sincerely hope you could kindly  revisit and potentially increase your score considering the efforts we have devoted during the discussion period.** We remain fully committed to addressing any further questions or suggestions you may have within the remaining discussion time and are grateful for your engagement throughout this process.

Thank you for your consideration.

Best regards,

The Authors

---

### Meta-Review · Area_Chair_Pfap · 2024-12-21

**Metareview:**

The paper introduces a method for improving the performance of GNNs with minimal hyperparameter tuning.  It propose a graph-conditioned generative model to generate (better) hyperparameter configurations based on lightly-tuned suboptimal parameters. This approach to parameter tuning is novel as far as I know. While the combination of techniques is compelling, the lack of a fundamentally new contribution limits the overall impact. The experimental evaluation is decent. Although some strongly-relevant baselines were missing, the authors added one such baseline in the rebuttal.

The biggest weakness of the paper is that the majority of the results are shown with "cluster" training, due to a lack of computational resources. Since, cluster training yields significantly worse performance than full training, the hyperparameters matter more and thus the proposed method looks better than it is. In response to the reviewers' request the authors included results with full training, where the gap to the baselines here is smaller. In my opinion this weakness is almost enough for a rejection, but I still recommend acceptance because the idea is interesting. However, I strongly advise the authors to include a prominent limitation section to highlight the issue.

**Additional Comments On Reviewer Discussion:**

Some reviewers (e.g., Reviewer TruH) noted insufficient justification of claims, and their concerns were not fully resolved during the rebuttal period. Reviewer 8AwD was initially critical of performance reporting and ablation studies but raised their score after additional experiments and clarifications. The authors conducted additional experiments (e.g., 120 with Bayesian optimization) and renamed the GAE module to avoid .

---

### Decision · Program_Chairs · 2025-01-22

Accept (Poster)